# CLIMB: Class-imbalanced Learning Benchmark on Tabular Data

**Zhining Liu**[1], **Zihao Li**[1], **Ze Yang**[1], **Tianxin Wei**[1], **Jian Kang**[2,3],
**Yada Zhu**[4], **Hendrik Hamann**[4,5,6], **Jingrui He**[1], **Hanghang Tong**[1]
[1]University of Illinois Urbana-Champaign [2] MBZUAI [3] University of Rochester
[4] MIT-IBM Watson AI Lab, IBM Research [5] Stony Brook University
[6] Brookhaven National Laboratory
liu326@illinois.edu

## Abstract

Class-imbalanced learning (CIL) on tabular data is important in many real-world applications where the minority class holds the critical but rare outcomes. In this paper, we present CLIMB, a comprehensive benchmark for class-imbalanced learning on tabular data. CLIMB includes 73 real-world datasets across diverse domains and imbalance levels, along with unified implementations of 29 representative CIL algorithms. Built on a high-quality open-source Python package with unified API designs, detailed documentation, and rigorous code quality controls, CLIMB supports easy implementation and comparison between different CIL algorithms. Through extensive experiments, we provide practical insights on method accuracy and efficiency, highlighting the limitations of naive rebalancing, the effectiveness of ensembles, and the importance of data quality. Our code, documentation, and examples are available at https://github.com/ZhiningLiu1998/imbalanced-ensemble.

## 1 Introduction

Class imbalance is a pervasive challenge in many real-world classification tasks, where the minority class often represents critical yet under-represented outcomes (He and Garcia, 2009; Johnson and Khoshgoftaar, 2019). Such challenges frequently arise in tabular data, which underpins many critical applications across industrial and scientific domains (Shwartz-Ziv and Armon, 2022), such as detecting fraud in financial transactions (Xiao et al., 2021), identifying malicious connections in network logs (Cieslak et al., 2006), and predicting positive diagnoses from medical records (Rahman and Davis, 2013). Given its significance in real-world decision-making, class-imbalanced learning (CIL) on tabular data has long been a key research focus in machine learning, AI and data mining.

However, the current landscape of benchmark resources for CIL on tabular data remains fragmented, with limited coverage across different algorithmic paradigms, datasets, and application domains. Most existing tabular benchmarks focus on orthogonal challenges such as distribution shift (Gardner et al., 2024), data augmentation (Machado et al., 2022), and adversarial robustness (Simonetto et al., 2024). Among the few benchmarks or empirical studies that address class-imbalanced tabular data, most focus narrowly on specific domains such as business (Zhu et al., 2018), finance (Xiao et al., 2021), healthcare (Khushi et al., 2021), or education (Wongvorachan et al., 2023), and the degree of imbalance tends to be similar. Moreover, these studies typically evaluate only a few methods within a single learning paradigm, lacking comprehensive comparisons across different types of CIL approaches (e.g., under/over-sampling, data cleaning, cost-sensitive, and their ensemble variants) in terms of both accuracy and efficiency. These limitations hinder a deeper understanding of how existing CIL methods perform on complex real-world tabular datasets with varying imbalance levels.

39th Conference on Neural Information Processing Systems (NeurIPS 2025) Track on Datasets and Benchmarks.

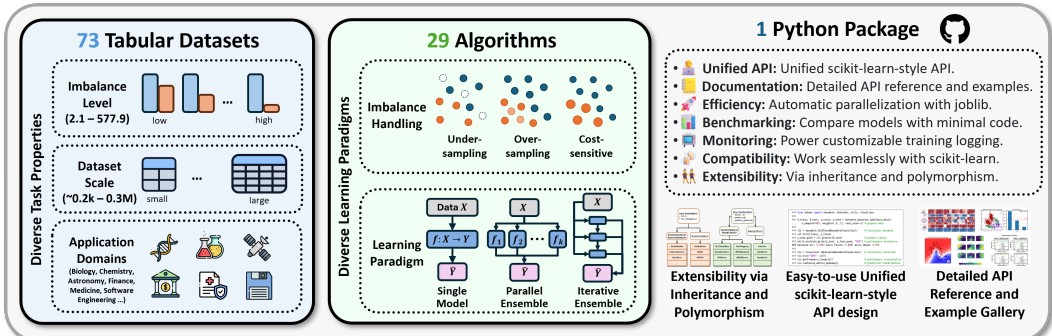

Figure 1: Overview of the proposed CLIMB benchmark. Best viewed in color.

To bridge this gap, we introduce CLIMB, a comprehensive benchmark for class-imbalanced learning on tabular data. CLIMB is based on our well-documented open-source Python package, which provides easy access to: **(1) a curated collection of 73 real-world tabular datasets** across diverse domains and imbalance levels, selected under rigorous criteria for non-triviality and realism, **(2) unified implementation of 29 representative CIL algorithms** covering resampling, cost-sensitive learning, and ensemble-based methods, **(3) principled benchmarking protocol** with comprehensive multi-fold data splits and hyperparameter searching to ensure fair comparisons. In addition, our library features: **(1) Unified API design:** we share and extend the unified API design of scikit-learn (Pedregosa et al., 2011) for ease-of-use and compatibility. **(2) Documentation and examples:** detailed API references, tutorials, and examples are provided; **(3) Quality assurance:** a suite of unit tests with 95% coverage is maintained and automatically executed through continuous integration; **(4) Easy extensibility:** algorithms are built with hierarchical and modularized abstractions, making it easy to incorporate new methods via inheritance and polymorphism. These components collectively establish CLIMB as a robust and user-friendly benchmark for class-imbalanced learning on tabular data. An overview of our CLIMB framework is provided in Figure 1.

Based on our benchmark, we have conducted extensive empirical experiments and analyses to assess the strengths and weaknesses of various CIL methods in terms of effectiveness, efficiency, and robustness. Our key takeaways are summarized as follows:

- **Class rebalancing is not always helpful.** In many cases, simple rebalancing techniques (including under-/over-sampling or cost-sensitive reweighting) tend to hurt rather than help classification performance, particularly under extreme imbalance scenarios.
- **Ensemble is critical for effective and robust CIL.** While rebalancing alone may be insufficient, combining it with ensemble strategies consistently leads to more accurate predictions and stable performance gain across different imbalance regimes.
- **Choose evaluation metrics wisely.** Different metrics emphasize different aspects of performance (e.g., AUPRC prioritizes minority class identification precision, while BAC is more sensitive to minority recall.) and may lead to different conclusions about model effectiveness.
- **Undersample ensembles strike a good performance-efficiency balance.** This paradigm is efficient due to (greatly) reduced training data and effective by combining diverse models trained on different subsets. This line of algorithms often matches or outperforms more costly competitors, thus a promising choice for large-scale or highly imbalanced scenarios.
- **Data quality matters, maybe more than class imbalance itself.** We find that adding 10% label noise or 30% missing features leads to a performance drop comparable to increasing the imbalance ratio by 500%. We believe this suggests that improving data quality may be as critical as, if not more than, solely addressing class imbalance in practice.

To summarize, our contributions in this work are three-fold: **(1) Comprehensive benchmark:** We introduce CLIMB, a general-purpose benchmark for class-imbalanced learning on tabular data. It includes a curated collection of 73 real-world datasets spanning diverse domains and imbalance levels, along with 29 representative CIL algorithms covering resampling, cost-sensitive learning, and ensemble-based approaches. **(2) High-quality open-source library:** We release a well-documented Python package that implements all benchmarked algorithms under a unified, extensible API. The library emphasizes usability, reliability, and extensibility, supported by our detailed documentation, rigorous code quality controls, and clean abstractions. **(3) Insights from extensive empirical analysis:** We perform large-scale experiments to evaluate the effectiveness, efficiency, and robustness

of existing CIL methods under class imbalance and noise. Our study reveals practical insights and failure modes, which we hope can guide future algorithm development and real-world deployment.

## 2 Related Works

Table 1: Comparison between this work and representative recent benchmark/empirical studies.

| Reference | Algorithm Coverage | | | | Dataset Coverage | | | Software Package |
|---|---|---|---|---|---|---|---|---|
| | Number | Resampling | Cost-sensitive | Ensemble | Number | Imbalance Ratio | Domain | |
| (Zhu et al., 2018) | 9 | ✓ | ✗ | ✗ | 11 | 5.9 - 54.6 | Business | ✗ |
| (Xiao et al., 2021) | 9 | ✓ | ✗ | ✗ | 6 | 1.3 - 28.1 | Finance | ✗ |
| (Khushi et al., 2021) | 21 | ✓ | ✗ | ✓ | 2 | 24.7 - 25.0 | Medical | ✗ |
| (Kim and Hwang, 2022) | 7 | ✓ | ✗ | ✗ | 31 | 1.1 - 577.9 | Multiple | ✗ |
| (Wongvorachan et al., 2023) | 4 | ✓ | ✗ | ✗ | 2 | 3.0 - 7.1 | Education | ✗ |
| **Ours** | **29** | ✓ | ✓ | ✓ | **73** | **2.1 - 577.9** | **Multiple** | ✓ |

**Class imbalance learning in different data modalities.** Class imbalance is prevalent in many real-world tasks where the class of interest contains rare but critical outcomes, such as financial fraud, network intrusions, or medical diagnoses (He and Garcia, 2009). These tasks frequently involve tabular data, a core modality in practical applications (Grinsztajn et al., 2022), and have been extensively studied over the past decades. This work focuses on the most popular data-level and algorithm-level CIL branches widely adopted in practice (Haixiang et al., 2017; Rezvani and Wang, 2023). We note that class imbalance is also a central concern in deep learning, efforts in that domain typically target structured data (e.g., images, text) through customized loss functions (Lin et al., 2017a) or architectural designs (Zhou et al., 2020). Since this line of work addresses an orthogonal set of challenges, we consider it outside the scope of this paper and refer interested readers to Johnson and Khoshgoftaar (2019); Ghosh et al. (2024) for comprehensive overviews of CIL in deep learning.

**Challenges of learning on imbalanced tabular data.** Unlike image and language data with natural structural priors, tabular data poses unique challenges such as heterogeneous feature types, small sample sizes, and the lack of meaningful local correlations (Grinsztajn et al., 2022). As a result, tree-based models remain the de facto choice for tabular tasks due to their robustness and inductive bias (Shwartz-Ziv and Armon, 2022), often outperforming deep learning methods. These challenges are further amplified under class imbalance, where limited samples in the minority class severely affect model generalization (Ghosh et al., 2024; Rezvani and Wang, 2023). Real-world tabular data also vary widely in scale and domain-specific patterns, complicating the search for universally effective CIL strategies. Our benchmark captures these factors by including datasets with diverse sizes, imbalance ratios, and domain complexities, and further introducing controllable noise and imbalance, enabling a comprehensive evaluation of how different CIL methods handle these challenges.

**Related benchmarks and empirical studies.** Most prior benchmarks on tabular data have centered on challenges that are largely independent of class imbalance, such as distribution shift (Gardner et al., 2024), data augmentation (Machado et al., 2022), and adversarial robustness (Simonetto et al., 2024). Only a handful of recent benchmarks or empirical investigations explicitly focused on class-imbalanced tabular learning, but they are typically restricted to specific application domains like business (Zhu et al., 2018), finance (Xiao et al., 2021), healthcare (Khushi et al., 2021), or education (Wongvorachan et al., 2023), often featuring datasets with comparable imbalance ratios. Additionally, these studies tend to explore a limited selection of algorithms confined to a single learning paradigm, which constrains their capacity to reveal comparative insights across diverse CIL techniques. In contrast, our work introduces a comprehensive benchmark that spans a broad spectrum of real-world tasks, varying imbalance levels, and algorithmic approaches. We highlight the differences between this work and representative related works in Table 1.

## 3 The CLIMB Benchmark

### 3.1 73 Reference Imbalanced Tabular Datasets

We compiled 73 naturally class-imbalanced tabular datasets provided by OpenML (Vanschoren et al., 2014) that span a wide real-world application domains with varying sizes and imbalance levels[1]. A statistical summary is provided in Figure 2. More detailed descriptions of each dataset can be found in Appendix A. They are selected using the following criteria:

---

[1]Access via: `https://imbalanced-ensemble.readthedocs.io/en/latest/api/datasets`

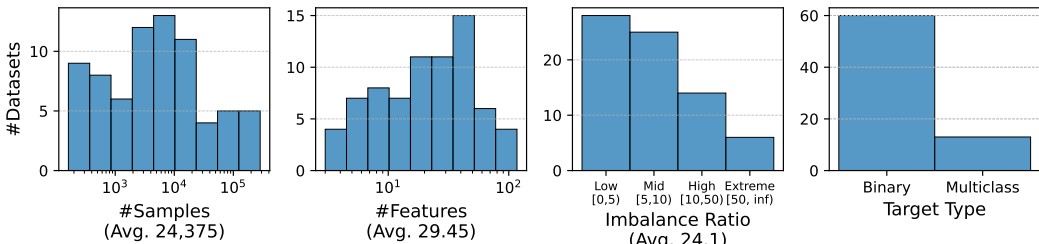

Figure 2: Statistics summary of the imbalanced tabular datasets included in CLIMB.

- **Real-world data & natural imbalance**: We select datasets collected from real-world scenarios, where the class distribution is naturally imbalanced. Artificially generated or manually imbalanced datasets are excluded to ensure that the evaluated tasks closely reflect practical applications.

- **Learning difficulty:** We discard datasets that are too easy to classify, i.e., we exclude those that can be nearly perfectly classified by a scikit-learn decision tree classifier, achieving an AUC-PR (a robust and informative metric for imbalanced classification) greater than 0.95.

- **Imbalance ratio:** Only datasets with an imbalance ratio (IR $:= \frac{\text{\#Majority Class}}{\text{\#Minority Class}}$) greater than 2 are retained. For multi-class datasets, we compute IR with the largest and smallest classes. We consider that datasets with even lower IR do not pose meaningful imbalance challenges for CIL and can typically be addressed by standard machine learning methods.

- **Data completeness:** We exclude datasets with missing values. This allows us to focus on the impact of class imbalance without introducing confounding factors related to missing data handling.

- **I.I.D. datasets**: We restrict our benchmark to datasets that follow the common i.i.d. assumption, thus excluding sequential or stream-based data such as time series.

- **Not Deterministic**: We remove datasets where the target is a deterministic function of the features, e.g., datasets on games like poker and chess. We believe that these datasets differ fundamentally from most real-world tabular problems and are better examined in separate benchmarks.

- **Undocumented datasets:** To ensure datasets are suitable for in-depth individual analysis, we exclude those lacking sufficient documentation. All selected datasets have reasonably detailed descriptions, either directly on OpenML or through referenced external sources.

## 3.2 29 Class-imbalanced Learning Algorithms

We implemented and evaluated 29 widely-used and highly-cited representative CIL algorithms. Each algorithm follows a standardized scikit-learn-style interface, accompanied by comprehensive documentation and usage examples. Based on their underlying mechanisms, these algorithms can be broadly categorized into the following groups:

- **Undersampling:** These methods balance classes by selecting a reduced set of majority samples, typically matching the minority class size. Techniques include Random Undersampling, Cluster Centroids (Lin et al., 2017b), Instance Hardness Threshold (Smith et al., 2014), and NearMiss (Mani and Zhang, 2003). While undersampling improves computational efficiency, it often comes at the cost of information loss due to the removal of many majority-class samples.

- **Cleaning:** Cleaning methods remove noisy or borderline majority samples to clarify decision boundaries for the minority class, typically using nearest-neighbor relationships. Examples include Tomek Links (Tomek, 1976b), Edited Nearest Neighbors (Wilson, 1972), Repeated ENN (Tomek, 1976a), AllKNN (Tomek, 1976a), One-Sided Selection (Kubat et al., 1997), and the Neighborhood Cleaning Rule (Laurikkala, 2001).

- **Oversampling:** Oversampling synthesizes new minority-class instances to balance the dataset. The most well-known method is SMOTE (Chawla et al., 2002), which creates synthetic samples via linear interpolation between a seed point and one of its nearest neighbors. Our benchmark includes its enhanced variants with targeted seed selection, such as Borderline-SMOTE (Han et al., 2005), SVM-SMOTE (Nguyen et al., 2011), ADASYN (He et al., 2008), and naive Random Oversampling. While preserving original data, oversampling may introduce unrealistic samples. They also significantly increase dataset size and leading to higher training cost.

- **Undersample Ensembles:** These methods ensemble multiple models trained on diverse under-sampled subsets, reducing information loss and improving robustness. Methods include Self-paced

Ensemble (Liu et al., 2020), Balance Cascade (Liu et al., 2008), Balanced Random Forest (Khoshgoftaar et al., 2007), EasyEnsemble (Liu et al., 2008), RUSBoost (Seiffert et al., 2009), and UnderBagging (Barandela et al., 2003). Beyond random undersampling, some methods leverage self-predictions to select informative subsets during training iteratively.

- **Oversample Ensembles:** These approaches build ensembles from multiple oversampled training sets, enhancing diversity without discarding data. Examples include OverBoost, SMOTE-Boost (Chawla et al., 2003), OverBagging, and SMOTEBagging (Wang and Yao, 2009). However, they are computationally the most expensive due to enlarged datasets and repeated training.

- **Cost-sensitive (Ensemble):** Cost-sensitive learning adjusts for imbalance by assigning higher misclassification costs to minority classes. We set costs inversely proportional to class frequencies. We also benchmark cost-sensitive ensemble variants including AdaCost (Fan et al., 1999), AdaUBoost (Karakoulas and Shawe-Taylor, 1998), and AsymBoost (Viola and Jones, 2001).

### 3.3 Benchmarking Protocol

**Dataset preprocessing.** We apply a unified preprocessing pipeline across all datasets to ensure consistent input formats and fair comparisons among algorithms. Specifically, all numerical features are standardized to have zero mean and unit variance. For categorical features, we adopt different encoding strategies based on their cardinality: binary categorical features (i.e., with only two unique values) are transformed using ordinal encoding into a single binary nominal feature, while those with more than two unique values are encoded using one-hot encoding.

**Data splitting.** To mitigate the randomness introduced by a single random train-test split, we adopt a 5-fold stratified splitting strategy for all datasets and report average performance. Specifically, each dataset is partitioned into five folds with the same class distributions (i.e., preserving the original class imbalance ratio). Each fold is used once as the test set while the remaining four folds are used for training. The final performance is reported as the average score across all splits.

**Algorithm configuration.** Given the strong performance and widespread use of tree-based models on tabular data and their close integration with certain CIL methods (e.g., Balanced Random Forest), we use decision trees as the base classifier to cooperate with all CIL algorithms. The ensemble size is set to 100 for all ensemble-based methods. To ensure fair and optimal evaluation, we perform hyperparameter tuning using Optuna (Akiba et al., 2019), with 100 optimization trials for each of the 23 CIL algorithms with tunable hyperparameters across all 73 datasets to determine the best-performing configurations. The search space and further details are provided in Appendix B.

**Evaluation metrics.** Classification accuracy is known to be misleading under class imbalance, as it is often dominated by the majority class(es) (He and Garcia, 2009). To provide a fair and balanced evaluation of model performance across both majority and minority classes, we adopt three widely used metrics: Area Under the Precision-Recall Curve (AUPRC), macro-averaged F1-score, and balanced accuracy. Among these, AUPRC evaluates model performance across varying classification thresholds and thus offers a more comprehensive assessment (Saito and Rehmsmeier, 2015).

## 4 Benchmark Results and Analysis

Following our rigorous benchmarking protocols, we conducted comprehensive experiments across all benchmark datasets to reveal insights into the classification performance, computational efficiency, and robustness of different CIL methods under varying levels of class imbalance. These experiments involved ∼0.8 million hyperparameter search trials, training of over 10 million base models, across 73 (datasets) × 30 (CIL methods) × 5 (splits) = 10,950 dataset-method-split pairs.

### 4.1 Main Benchmark Results

We report the main benchmark results in Table 2. To better present insights from the large volume of numerical results, we grouped the 73 datasets by imbalance ratio (IR) into four categories: low (IR< 5), medium (IR∈ [5, 10)), high (IR∈ [10, 50)), and extreme (IR> 50) imbalance. We report the performance and ranking of each CIL method averaged over each dataset within each group.

**RQ1: Balancing or Cleaning?** Table 2 shows that rebalancing-based CIL methods (including undersampling, oversampling, and cost-sensitive approaches) often lead to performance degradation instead of gains compared to no balancing (highlighted by red cells). Undersampling causes notable drops in AUPRC and F1 even on low-imbalance datasets due to information loss. Oversampling and cost-sensitive show degradation on highly imbalanced datasets, suggesting that synthesizing minority

Table 2: Main benchmark results. Given the large number of results, we group the 73 datasets by imbalance level into 4 categories and report the averaged AUPRC (AP), macro F1, and Balanced Accuracy (BAC) for each CIL method (in $\times 10^{-2}$). Detailed results for each dataset can be found in C. For a comprehensive evaluation, we also rank all methods on each dataset and metric, and report their average ranks. **Color coding is used to show the performance gains (blue) or losses (red) relative to the base no-balancing method, with deeper colors indicating larger differences.**

| Dataset Group | Avg. Stat | Metric | Base | Undersample |  |  |  | Cleaning |  |  |  |  |  | Oversample |  |  |  |  | Undersample Ensemble |  |  |  |  |  | Oversample Ensemble |  |  |  | Cost-sensitive |  |  |  |
|---|---|---|---|---|---|---|---|---|---|---|---|---|---|---|---|---|---|---|---|---|---|---|---|---|---|---|---|---|---|---|---|---|
| | | | | RUS | CC | IHT | NM | TL | ENN | RENN | AKNN | OSS | NCR | ROS | SMT | BSMT | SSMT | ASYN | SPE | BC | BRF | EE | UBS | UBA | OBS | SMBS | OBA | SMBA | CS | AdaC | AdaBS | AsyBS |
| IR ∈ (0, 5) (28 datasets) | Score (↑) | AP | 51.0 | 49.4 | 48.0 | 45.8 | 45.5 | 51.5 | 53.6 | 53.4 | 52.9 | 51.7 | 53.7 | 51.1 | 51.6 | 51.8 | 52.1 | 51.6 | 59.3 | 57.8 | 57.7 | 59.0 | 58.6 | 59.0 | 52.9 | 53.7 | 58.4 | 58.9 | 51.2 | 52.5 | 52.4 | 52.5 |
| | | F1 | 72.0 | 70.4 | 67.1 | 66.0 | 65.5 | 72.8 | 73.9 | 73.3 | 73.0 | 73.0 | 74.2 | 72.4 | 72.9 | 72.9 | 73.2 | 72.8 | 77.9 | 76.8 | 76.7 | 78.0 | 77.5 | 78.0 | 73.7 | 74.3 | 76.5 | 77.4 | 72.3 | 73.2 | 73.2 | 73.2 |
| | | BAC | 72.1 | 73.3 | 71.7 | 73.8 | 69.7 | 73.3 | 76.5 | 76.7 | 76.4 | 73.6 | 76.6 | 72.5 | 73.5 | 73.6 | 73.8 | 73.6 | 78.5 | 77.3 | 79.7 | 79.7 | 78.7 | 79.7 | 73.7 | 74.8 | 75.7 | 77.0 | 72.4 | 73.2 | 73.1 | 73.2 |
| | Rank (↓) | AP | 21.2 | 25.0 | 26.9 | 23.1 | 26.3 | 18.9 | 12.7 | 13.7 | 15.6 | 18.2 | 12.9 | 21.5 | 20.0 | 19.5 | 18.9 | 20.1 | 4.5 | 8.0 | 7.5 | 5.3 | 5.4 | 5.1 | 16.5 | 14.3 | 6.1 | 4.6 | 21.9 | 17.0 | 17.6 | 16.4 |
| | | F1 | 20.6 | 26.1 | 27.3 | 28.3 | 27.5 | 17.8 | 13.8 | 15.9 | 17.8 | 16.7 | 13.2 | 20.5 | 19.0 | 18.5 | 17.2 | 18.8 | 3.8 | 7.1 | 8.3 | 4.6 | 5.3 | 4.3 | 15.8 | 13.0 | 8.2 | 5.1 | 21.3 | 16.1 | 17.5 | 15.5 |
| | | BAC | 24.3 | 20.3 | 23.9 | 19.3 | 25.4 | 19.2 | 11.2 | 10.8 | 12.1 | 17.9 | 10.8 | 23.6 | 20.0 | 19.1 | 17.5 | 18.9 | 6.3 | 8.7 | 3.5 | 3.5 | 5.8 | 3.4 | 19.7 | 14.3 | 11.9 | 8.8 | 24.8 | 19.6 | 21.3 | 19.2 |
| IR ∈ [5, 10) (24 datasets) | Score (↑) | AP | 50.9 | 43.2 | 35.3 | 40.0 | 32.1 | 51.0 | 52.5 | 52.6 | 52.5 | 50.8 | 52.5 | 51.1 | 51.7 | 51.4 | 51.9 | 50.7 | 64.6 | 62.7 | 60.5 | 62.4 | 63.8 | 62.4 | 54.8 | 54.1 | 61.2 | 62.8 | 51.4 | 54.1 | 54.4 | 54.1 |
| | | F1 | 74.7 | 68.4 | 56.7 | 64.2 | 57.4 | 74.6 | 75.1 | 75.0 | 75.0 | 74.5 | 75.2 | 74.4 | 75.1 | 75.0 | 75.3 | 74.6 | 79.7 | 78.7 | 76.7 | 78.7 | 79.3 | 78.7 | 75.6 | 75.8 | 77.0 | 78.5 | 74.5 | 75.4 | 75.4 | 75.4 |
| | | BAC | 74.7 | 76.0 | 71.2 | 76.5 | 70.8 | 74.9 | 77.6 | 78.0 | 77.7 | 74.8 | 77.8 | 74.4 | 76.4 | 75.8 | 76.4 | 75.7 | 82.1 | 80.8 | 83.1 | 82.9 | 82.5 | 82.9 | 75.4 | 76.8 | 75.6 | 77.7 | 74.5 | 75.0 | 75.1 | 75.0 |
| | Rank (↓) | AP | 20.6 | 25.3 | 28.6 | 25.2 | 28.6 | 20.2 | 14.9 | 14.3 | 15.4 | 19.9 | 15.0 | 20.3 | 17.8 | 18.7 | 17.1 | 20.0 | 2.6 | 5.8 | 8.5 | 6.0 | 3.3 | 5.8 | 14.2 | 15.1 | 9.4 | 6.5 | 20.0 | 15.5 | 15.4 | 15.1 |
| | | F1 | 18.8 | 27.1 | 29.2 | 28.3 | 29.2 | 18.5 | 15.6 | 15.6 | 15.7 | 18.6 | 15.3 | 18.8 | 16.5 | 17.6 | 15.2 | 18.8 | 3.7 | 6.2 | 12.7 | 6.4 | 3.5 | 6.0 | 14.2 | 14.3 | 10.5 | 5.8 | 18.4 | 15.0 | 15.2 | 14.4 |
| | | BAC | 22.4 | 18.2 | 25.2 | 17.3 | 25.5 | 21.7 | 14.1 | 12.9 | 13.8 | 20.8 | 13.7 | 22.4 | 14.7 | 18.2 | 14.5 | 18.5 | 5.3 | 6.8 | 2.9 | 3.6 | 3.6 | 3.1 | 18.8 | 14.2 | 17.1 | 11.8 | 22.2 | 20.6 | 20.5 | 20.5 |
| IR ∈ [10, 50) (15 datasets) | Score (↑) | AP | 34.9 | 23.6 | 17.4 | 27.1 | 14.5 | 35.1 | 36.2 | 36.2 | 35.9 | 35.5 | 36.3 | 34.1 | 34.3 | 35.8 | 35.0 | 34.1 | 47.1 | 41.4 | 38.4 | 41.9 | 44.9 | 41.9 | 36.7 | 36.6 | 45.0 | 46.0 | 34.1 | 36.4 | 36.7 | 36.4 |
| | | F1 | 61.6 | 51.2 | 35.8 | 52.9 | 35.6 | 61.6 | 62.3 | 61.9 | 61.5 | 61.9 | 62.4 | 61.0 | 61.0 | 62.1 | 61.7 | 61.0 | 65.5 | 62.1 | 59.7 | 62.9 | 64.0 | 62.9 | 61.6 | 61.7 | 64.0 | 65.8 | 61.0 | 61.5 | 61.6 | 61.5 |
| | | BAC | 61.8 | 63.6 | 56.7 | 65.8 | 54.3 | 62.1 | 64.5 | 65.0 | 64.3 | 62.3 | 64.1 | 60.8 | 63.5 | 63.7 | 63.5 | 63.2 | 70.7 | 67.8 | 72.2 | 72.1 | 70.4 | 72.1 | 61.1 | 64.0 | 62.4 | 65.1 | 61.0 | 61.5 | 61.2 | 61.5 |
| | Rank (↓) | AP | 18.4 | 25.5 | 29.4 | 21.5 | 29.4 | 17.9 | 13.3 | 13.7 | 14.5 | 16.3 | 12.7 | 20.9 | 19.8 | 15.3 | 17.4 | 19.7 | 3.9 | 8.9 | 13.3 | 9.4 | 5.4 | 9.2 | 14.0 | 14.6 | 7.4 | 5.5 | 20.6 | 16.7 | 14.5 | 16.2 |
| | | F1 | 14.2 | 27.9 | 29.5 | 26.9 | 29.4 | 15.2 | 10.6 | 12.0 | 14.3 | 13.0 | 11.5 | 18.7 | 18.4 | 12.3 | 15.1 | 18.4 | 4.8 | 14.3 | 18.9 | 11.5 | 7.5 | 11.3 | 14.5 | 15.5 | 10.0 | 5.7 | 18.0 | 15.7 | 14.9 | 15.1 |
| | | BAC | 20.2 | 16.8 | 27.2 | 11.9 | 28.0 | 19.9 | 14.7 | 13.0 | 15.4 | 18.6 | 14.2 | 24.1 | 14.7 | 13.1 | 14.7 | 16.7 | 5.3 | 6.9 | 2.9 | 2.9 | 5.5 | 2.1 | 22.7 | 14.0 | 17.9 | 13.8 | 23.0 | 21.1 | 22.6 | 21.0 |
| IR ∈ [50, 1000) (6 datasets) | Score (↑) | AP | 42.6 | 18.9 | 15.9 | 33.0 | 13.5 | 44.2 | 45.0 | 44.1 | 44.9 | 44.3 | 44.6 | 41.7 | 37.1 | 42.2 | 41.9 | 34.3 | 57.5 | 50.1 | 32.9 | 35.5 | 43.3 | 35.5 | 45.0 | 40.5 | 56.4 | 56.0 | 41.7 | 48.1 | 46.6 | 48.1 |
| | | F1 | 74.0 | 50.6 | 35.2 | 68.0 | 32.9 | 75.0 | 75.1 | 74.7 | 74.8 | 75.1 | 75.0 | 73.6 | 71.9 | 74.4 | 73.7 | 70.3 | 74.7 | 68.6 | 56.3 | 59.8 | 68.5 | 59.8 | 74.3 | 71.9 | 75.8 | 76.6 | 73.9 | 74.9 | 74.9 | 74.9 |
| | | BAC | 74.6 | 81.8 | 70.5 | 79.9 | 66.2 | 75.1 | 75.8 | 76.1 | 75.6 | 75.2 | 75.6 | 73.1 | 77.2 | 76.1 | 76.0 | 76.3 | 85.9 | 83.0 | 88.0 | 87.3 | 85.7 | 87.3 | 73.5 | 77.6 | 72.4 | 74.4 | 73.1 | 74.8 | 73.2 | 74.8 |
| | Rank (↓) | AP | 16.7 | 27.7 | 29.2 | 20.5 | 29.8 | 15.8 | 12.3 | 13.3 | 13.2 | 14.3 | 15.0 | 18.5 | 18.3 | 15.3 | 16.8 | 21.7 | 2.7 | 8.5 | 17.5 | 15.3 | 9.7 | 15.0 | 14.2 | 16.7 | 7.3 | 6.3 | 18.5 | 11.0 | 13.5 | 10.3 |
| | | F1 | 13.2 | 27.8 | 29.5 | 18.8 | 29.5 | 11.0 | 9.0 | 11.5 | 11.0 | 9.8 | 11.2 | 15.3 | 14.7 | 11.8 | 13.8 | 18.5 | 10.8 | 15.8 | 26.8 | 24.7 | 18.3 | 25.7 | 12.5 | 13.8 | 9.5 | 7.8 | 13.3 | 9.5 | 11.3 | 8.5 |
| | | BAC | 20.8 | 7.3 | 21.8 | 10.3 | 23.2 | 20.3 | 13.7 | 14.5 | 14.7 | 19.3 | 15.7 | 25.2 | 13.7 | 16.2 | 16.0 | 17.3 | 4.7 | 5.3 | 2.2 | 3.0 | 4.3 | 2.7 | 23.5 | 12.5 | 26.0 | 21.5 | 24.3 | 20.5 | 24.7 | 19.8 |

**\*Abbreviations:** Random Undersampling (**RUS**), Cluster Centroids (**CC**), Instance Hardness Threshold (**IHT**), NearMiss (**NM**), Tomek Links (**TL**), Edited Nearest Neighbors (**ENN**), Repeated ENN (**RENN**), AllKNN (**AKNN**), One-Sided Selection (**OSS**), Neighborhood Cleaning Rule (**NCR**), Random Oversampling (**ROS**), SMOTE (**SMT**), Borderline SMOTE (**BSMT**), SVM SMOTE (**SSMT**), ADASYN (**ASYN**), Self-paced Ensemble (**SPE**), Balance Cascade (**BC**), Balanced Random Forest (**BRF**), Easy Ensemble (**EE**), RUSBoost (**UBS**), UnderBagging (**UBA**), OverBoost (**OBS**), SMOTEBoost (**SMBS**), OverBagging (**OBA**), SMOTEBagging (**SMBA**), Cost-sensitive (**CS**), AdaCost (**AdaC**), AdaUBoost (**AdaBS**), AsymBoost (**AsyBS**).

samples and reweighting are not robust when the minority class is poorly represented. In contrast, cleaning methods with less aggressive data modifications demonstrate more stable performance.

> **Takeaway #1: Class rebalancing is not always helpful, while cleaning can be a safer choice.**
>
> *When used alone, focusing on preserving or improving representation quality through cleaning seems to be a safer and more robust strategy than balanced resampling or reweighting.*

**RQ2: Does ensemble help?** The top-performing methods (highlighted by blue cells) across different imbalance levels and metrics are predominantly ensemble-based. Interestingly, while standalone undersampling methods perform poorly, undersample ensembles effectively mitigate information loss by combining multiple views and lead to strong results. Even simple approaches based on random undersampling (e.g., BRF, UBS, UBA) perform well under low to medium imbalance. For highly imbalanced cases, methods like SPE and BC further improve by leveraging self-predictions to iteratively select informative subsets, achieving performance comparable to more expensive oversample ensembles. Among the oversample ensembles, Bagging-based methods (OBA, SMBA) perform better than Boosting-based ones (OBS, SMBS), especially in highly-imbalanced cases. We attribute this to the introduction of low-quality and hard-to-classify synthetic samples by oversampling strategies like SMOTE: boosting-based methods may overemphasize these low-quality synthetic samples, while bagging is generally more robust to noise within the dataset.

> **Takeaway #2: Ensemble is a critical technique for effective and robust CIL.**
>
> *Ensembles achieve balanced and robust learning by aggregating multiple rebalanced views. They mitigate information loss from undersampling, enhance diversity from oversampling, and consistently outperform single models across all imbalance levels.*

**RQ3: How to select evaluation metric(s)?** We also note that different metrics may lead to different conclusions. For instance, in the extreme imbalance (IR>50) group, RUS and its ensembles (e.g., BRF, UBA) typically improve BAC but degrade AUPRC and F1, whereas oversampling and cost-sensitive ensembles (e.g., OBA, AdaBS) show the opposite trend. This reflects the different focus of each metric (Jeni et al., 2013; Japkowicz, 2013): AUPRC and F1 prioritize precision and accurate minority class identification, making them sensitive to false positives, while BAC emphasizes balanced recall across classes. Undersampling improves minority recall by discarding most majority samples but at the cost of precision (misclassifying many majority instances), whereas oversampling and cost-sensitive methods better preserve precision, sometimes sacrificing the recall of minority samples. In practice, metric choice should be informed by domain knowledge, e.g., precision is critical in spam detection to avoid misclassifying legitimate emails and disrupting user communication, while recall is paramount in cancer screening to prevent missing true cases (Haixiang et al., 2017).

---

**Takeaway #3: Different metrics may lead to different conclusions for certain methods.**

*Different metrics emphasize different aspects of performance evaluation and sometimes lead to different conclusions. In practice, one should choose appropriate metrics based on domain needs for a more accurate interpretation of model effectiveness.*

---

### 4.2 Performance versus Runtime Analysis

**Setup.** Beyond classification performance, the runtime efficiency of CIL algorithms is also crucial for practical applications. Figure 3 presents a performance versus runtime analysis to illustrate the utility-efficiency trade-off of different models across dataset groups with different imbalance levels. Runtimes were measured on a workstation with an Intel Core i9 12900 CPU.

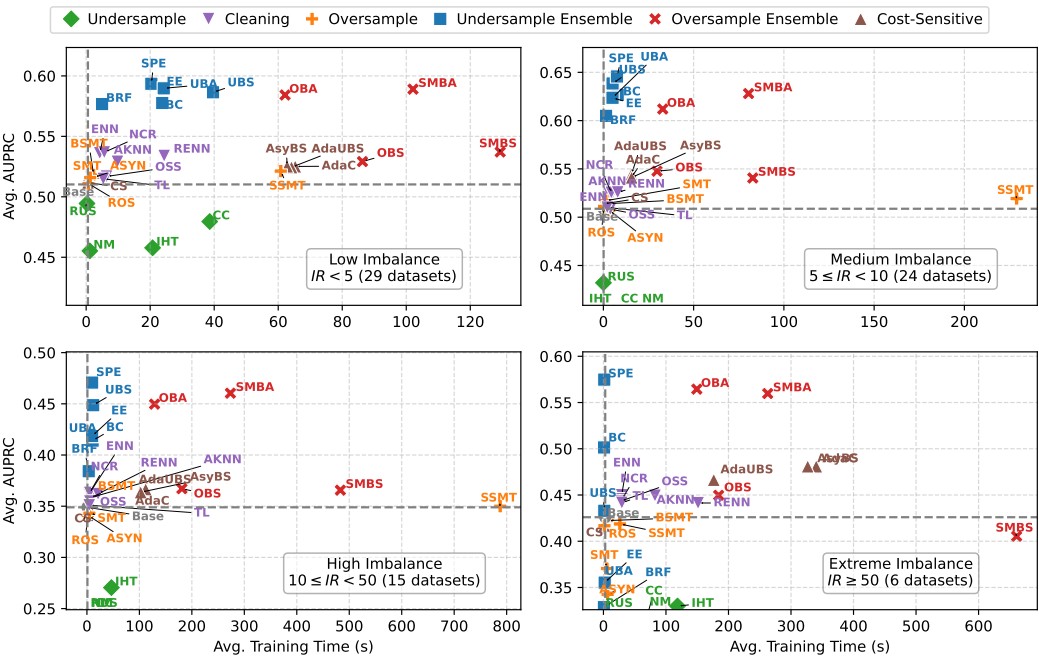

Figure 3: Performance versus runtime analysis, following the dataset grouping in Table 2. The **x-axis** shows the average runtime of each CIL algorithm, and the **y-axis** shows the average AUPRC. **Desired methods are closer to the upper-left corner with high accuracy and low computational cost.** Different markers indicate different CIL method categories, the dashed line denotes the base model (no balancing) performance and runtime. More results with other metrics are in Appendix C.

**RQ4: Which (types of) methods are costly and why?** **(i)** *For non-ensemble methods,* cost differences mainly arise from the overhead of the resampling operation itself, while the impact of training sample size is relatively minor. For example, complex undersampling methods (e.g., clustering-based CC and probability-based IHT) tend to be more time-consuming than simpler oversampling approaches. **(ii)** *For ensemble methods,* cost differences are primarily driven by the

*size of the training data and the ensemble training paradigm.* Runtime cost generally follows the order: undersampling < cost-sensitive < oversampling, and bagging-based < boosting-based. The reason behind is that ensemble methods typically do not rely on overly complex balancing operations, but training multiple base models significantly amplifies the impact of training set size and ensemble strategy on the overall runtime. **(iii)** ***Other remarks:*** We note that the runtime observations are not comparable between different dataset groups as their datasets vary in size and dimensions. Also, the importance of training set size in runtime may change if we use base models that are more/less sensitive to dataset scale.

**RQ5: Are ensemble methods always more expensive to train?** *Not necessarily.* For instance, complex undersampling methods like IHT and CC are often slower than many undersample ensembles, even though the latter needs to train 100 base models. Similarly, SVM-SMOTE (SSMT), which requires training an auxiliary SVM model for oversampling, can in some cases be more time-consuming than all tested tree-based ensembles. Notably, we observe that undersample ensembles often achieve strong predictive performance with relatively low computational cost. Even under extreme imbalance, iterative informed undersampling variants such as SPE and BC continue to perform robustly.

> **Takeaway #4: Undersample ensembles strike a good accuracy-efficiency balance.**
>
> *Undersample ensembles deliver strong results at low cost by reducing training data and aggregating diverse views. The best variants often rival or outperform more expensive counterparts.*

### 4.3 Robustness Analysis

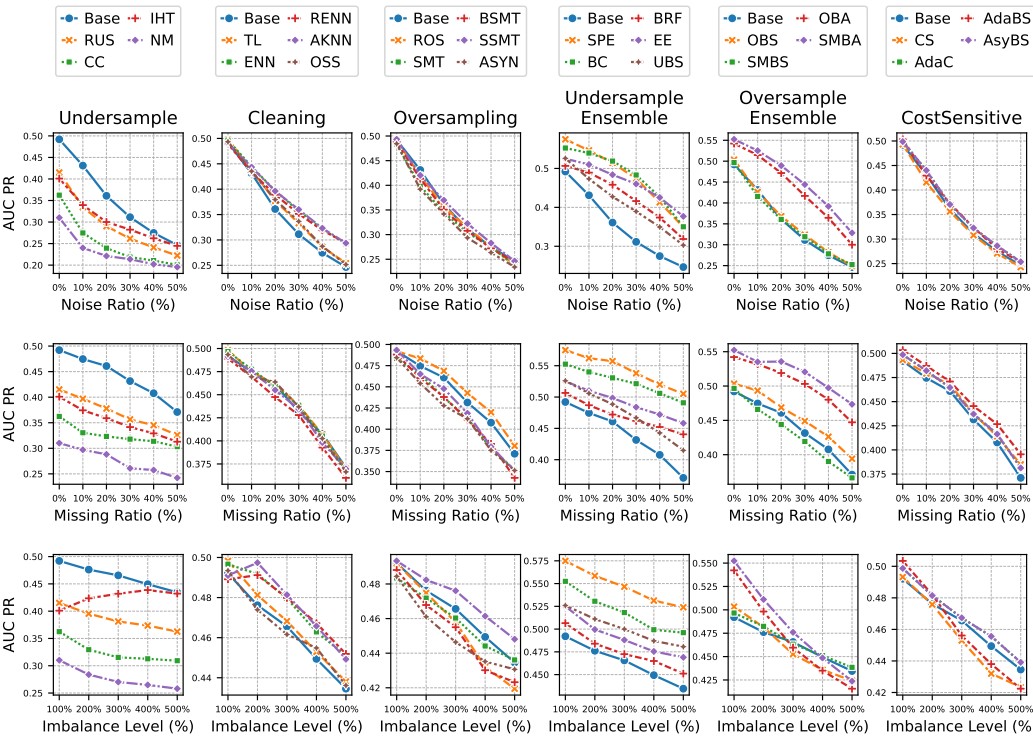

Figure 4: Robustness analysis. Each row corresponds to the noise, missing values, and additional class imbalance setting (from top to bottom). Each column represents a branch of CIL methods.

**Setup.** Finally, we conduct controlled experiments to study how noise, missing values, and more severe class imbalance impact CIL model performance, offering insights for handling similar difficulties in practical applications. To ensure a fair comparison, each factor is introduced individually while keeping other factors unchanged. **(i) Label noise:** We introduce label flipping noise to simulate real-world annotation errors. The noise ratio is defined on minority class, e.g., a 10% noise ratio means that 10% of the minority-class samples are randomly relabeled as other classes, while an equal number of non-minority samples are relabeled as the minority class. This preserves the original IR. **(ii)**

**Missing value:** Given the missing ratio, we randomly mask corresponding number of values across all samples and features, replace them with the mean value observed for each respective feature. This setting simulates the common practice of mean imputation in real-world applications. **(iii) Additional imbalance:** We intensify class imbalance by further removing samples from the minority class. For example, a 200% imbalance level means that 50% of the minority-class samples are removed, thus doubling the original IR. Figure 4 shows the results averaged over all tested datasets.

**RQ6: Are CIL methods robust to additional difficulty factors?** *Generally, yes.* In most cases, the relative gain or loss of CIL methods compared to the base (no-balancing) setting remains consistent across different levels of noise, missing values, and additional class imbalance. The ranking among CIL methods also remains largely stable. A few exceptions: (i) IHT shows improvement as imbalance increases. This is because IHT removes hard examples that classifiers do not predict confidently. As minority class shrinks, such hard examples become fewer, causing IHT to gradually degenerate toward no-balancing behavior. (ii) OBA and SMBA show huge performance drops under extreme imbalance. We attribute this to the further reduction in minority-class size, which limits the ability of oversampling and synthetic samples to enhance minority-class representation.

**RQ7: Which factor has a greater impact?** Interestingly, we observe that noise and missing values have a greater impact on model performance than class imbalance. For the base model, introducing 10% label noise or 30% missing features results in a similar performance drop of increasing the imbalance ratio by 500%. This implies the importance of maintaining data quality, which also aligns with our earlier finding on the effectiveness of data cleaning methods, as discussed in Takeaway #1.

> **Takeaway #5: Data quality greatly affects CIL, if not more than class imbalance itself.**
>
> *Noisy labels and missing features can degrade model performance as much as, or even more than, severe class imbalance. Ensuring high data quality is crucial for building robust models and should be prioritized alongside class rebalancing.*

**Additional results in the appendix.** Due to space limitations, we present the key results and insights in the main text. Appendix C includes results with **hybrid sampling methods and GBDTs, pairwise win-ratio comparisons, full per-dataset evaluation scores, and runtime analyses.**

## 5 Conclusion and Future Directions

**Limitations and Future Directions.** While we have conducted a comprehensive study given available resources, many interesting questions remain open for future work. Building on our findings, we highlight several promising directions to further extend our work and advance the field of CIL:

- Conducting similar analyses under the combined effects of class imbalance and other data quality challenges, such as noise, missing values, class overlapping (Santos et al., 2022), and small disjuncts (Jo and Japkowicz, 2004). This may be facilitated by developing flexible, realistic tabular data synthesis frameworks (Liu et al., 2024).
- Investigating the effectiveness of deep learning-based solutions. Although tree-based models generally outperform deep models on tabular data (Grinsztajn et al., 2022), combining deep architectures with established CIL paradigms (e.g., undersample ensembles) and other forms of inductive bias may enable more effective deep imbalanced learning.
- Examining the integration of CIL methods with non-tree-based models to explore whether different types of base learners provide unique advantages on imbalanced tabular data.
- Exploring combinations of different CIL paradigms, such as dynamically integrating data cleaning into ensembles to enhance robustness against low-quality data. Additionally, designing AutoML systems that can automatically compose these modules during inference presents an interesting future direction (Barbudo et al., 2023; Karmaker et al., 2021).

**Conclusion.** In this paper, we introduced CLIMB, a comprehensive benchmark for class-imbalanced learning (CIL) on tabular data. CLIMB provides a curated collection of 73 real-world datasets spanning diverse domains and imbalance levels, along with unified implementations of 29 representative CIL algorithms. Built upon a high-quality open-source library, CLIMB enables fair, reproducible, and extensible evaluation of CIL methods. Through empirical studies involving millions of model trainings and hyperparameter searches, we drew several practical insights. (i) naïve class rebalancing alone is

often ineffective, while data cleaning offers a safer improvement strategy; (ii) ensemble methods are critical for robust and effective CIL; (iii) the evaluation metric may affect the conclusion and should be chosen wisely; (iv) undersample ensembles strike a favorable balance between performance and efficiency; (v) data quality issues, such as label noise and missing values, can have even greater impact on model performance than class imbalance itself. We hope that CLIMB will serve as a solid foundation for advancing future research on class-imbalanced learning and promote the development of more reliable and practical solutions for real-world challenges.

## Acknowledgments and Disclosure of Funding

This work is supported by the National Science Foundation (IIS-2117902 and 2433308), MIT-IBM Watson AI Lab, and IBM-Illinois Discovery Accelerator Institute. The views and conclusions are those of the authors and should not be interpreted as representing the official policies of the funding agencies or the government. The U.S. Government is authorized to reproduce and distribute reprints for Government purposes, notwithstanding any copyright notation here on.

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

# Appendix

This appendix provides additional details to support the experiments and findings presented in the main paper. Section A presents details of the datasets used in our benchmark. Section B describes further reproducibility details such as hyperparameter search strategy and runtime measurement protocols Finally, Section C provides extended results and analyses, including additional CIL baselines (hybrid sampling methods and GBDTs), pairwise win-ratio comparisons, discussion on the advantages of Self-paced Ensemble, comparison with the BAF benchmark, and full per-dataset evaluation scores and runtime statistics.

# A    Datasets Details

**Dataset descriptions.**    The 73 datasets we evaluated span a wide range of imbalance levels, sizes, and dimensions, and were selected based on the seven rigorous criteria outlined in Section 3.1. All datasets were collected from real-world scenarios and naturally exhibit class imbalance. We note that OpenML includes numerous artificially generated imbalanced datasets, but these were excluded to ensure that the evaluated tasks closely reflect practical applications. The final collection covers tasks from diverse real-world domains such as finance, medicine, and engineering. Table 3 summarizes key information for each dataset, including name, number of samples and features, imbalance ratio, target type, domain, and a brief task description. Due to the large number of datasets, we do not provide individual citations here. Full dataset descriptions and reference publications can be found on their respective OpenML pages or in the referenced external sources therein.

Table 3: Dataset statistics and descriptions.

| Dataset | #Samples | #Features | IR | Type | Domain | Description |
|---|---|---|---|---|---|---|
| bwin_amlb | 530 | 13 | 2.01 | Binary | Behavioral Analytics | Aggregated data on virtual and live sports betting behavior over a multi-month period. |
| mozilla4 | 15545 | 5 | 2.04 | Binary | Software Engineering | Tracks defect fixes and code size changes in Mozilla C++ classes over time. |
| mc2 | 161 | 39 | 2.1 | Binary | Software Engineering | NASA software defect data using McCabe and Halstead complexity metrics. |
| vertebra-column | 310 | 6 | 2.1 | Binary | Medicine | Biomechanical features used to classify vertebral column pathologies. |
| wholesale-customers | 440 | 8 | 2.1 | Binary | Retail | Annual spending profiles of wholesale distribution customers across product categories. |
| law-school-admission-bianry | 20800 | 14 | 2.11 | Binary | Education | Binary prediction of law school applicants' UGPA with demographic attributes. |
| bank32nh | 8192 | 32 | 2.22 | Binary | Finance | Bank dataset with a binarized target based on mean threshold. |
| elevators | 16599 | 18 | 2.24 | Binary | Robotics | Control application data by thresholding numeric targets. |
| cpu_small | 8192 | 12 | 2.31 | Binary | Computer Systems | Binarized CPU performance data from original regression targets. |
| Credit_Approval_Classification | 1000 | 50 | 2.33 | Binary | Finance | Predicts credit approval based on demographic and financial features. |
| house_8L | 22784 | 8 | 2.38 | Binary | Real Estate | House price data with binarized target values based on average threshold. |
| house_16H | 22784 | 16 | 2.38 | Binary | Real Estate | Higher-dimensional version of house price data with binarized targets. |
| phoneme | 5404 | 5 | 2.41 | Binary | Speech Recognition | Classification of nasal vs. oral phonemes using harmonic amplitude features. |
| ilpd-numeric | 583 | 10 | 2.49 | Binary | Medicine | Liver disorder classification with all-numeric features. |
| planning-relax | 182 | 12 | 2.5 | Binary | Neuroscience | EEG signal data distinguishing planning vs. relaxation mental states. |
| MiniBooNE | 130064 | 50 | 2.56 | Binary | Physics | Distinguishes electron from muon neutrinos in a particle experiment. |
| machine_cpu | 209 | 6 | 2.73 | Binary | Computer Systems | Binarized CPU benchmark dataset based on performance metrics. |
| telco-customer-churn | 7043 | 39 | 2.77 | Binary | Business | Telecom customer churn prediction based on service and usage data. |
| haberman | 306 | 3 | 2.78 | Binary | Medicine | Survival analysis of breast cancer patients after surgery. |
| vehicle | 846 | 18 | 2.88 | Binary | Automotive | Binarized vehicle type classification dataset based on majority class. |
| cpu | 209 | 36 | 2.94 | Binary | Computer Systems | CPU performance data converted into binary classification task. |
| ada | 4147 | 48 | 3.03 | Binary | Sociology | Discover high revenue people from census data. |
| adult | 48842 | 107 | 3.18 | Binary | Sociology | Predicts income level (>50K) from census features. |
| blood-transfusion-service-center | 748 | 4 | 3.2 | Binary | Health | Predicts blood donation behavior based on RFM features. |
| default-of-credit-card-clients | 30000 | 23 | 3.52 | Binary | Finance | Predicts default risk for credit card clients based on payment and bill history. |
| Customer_Churn_Classification | 175028 | 24 | 3.74 | Binary | Business | Predicts customer churn based on service usage and demographics. |
| SPECTF | 267 | 44 | 3.85 | Binary | Medicine | Diagnoses cardiac conditions from SPECT imaging features. |
| Medical-Appointment-No-Shows | 110527 | 36 | 3.95 | Binary | Healthcare | Predicts patient no-shows for medical appointments based on demographics and history. |
| JapaneseVowels | 9961 | 14 | 5.17 | Binary | Speech Recognition | Binarized classification of speaker voice samples originally from a multi-class dataset. |
| ibm-employee-attrition | 1470 | 53 | 5.2 | Binary | Human Resources | Predicts employee attrition based on job satisfaction and personal features. |
| first-order-theorem-proving | 6118 | 51 | 5.26 | Multiclass | Automated Reasoning | Feature-based dataset for learning heuristics in first-order theorem proving. |
| user-knowledge | 403 | 5 | 5.38 | Multiclass | Education | Models students' domain knowledge in electrical machines based on performance and behavior. |
| online-shoppers-intention | 12330 | 28 | 5.46 | Binary | E-commerce | Predicts purchase intention based on session behavior and web metrics. |
| kc1 | 2109 | 21 | 5.47 | Binary | Software Engineering | NASA defect prediction dataset with code complexity metrics. |
| thoracic-surgery | 470 | 16 | 5.71 | Binary | Medicine | Predicts 1-year survival after lung cancer surgery. |
| UCI_churn | 3333 | 14 | 5.9 | Binary | Business | Customer churn prediction dataset with limited metadata. |
| arsenic-female-bladder | 559 | 4 | 5.99 | Binary | Medicine | Binarized dataset likely related to bladder health outcomes in females with arsenic exposure. |
| okcupid_stem | 26677 | 117 | 6.83 | Multiclass | Sociology | Profiles from OkCupid used to predict whether a user has a STEM-related job. |
| ecoli | 327 | 7 | 7.15 | Multiclass | Biology | Studies the cellular localization sites of E. coli proteins. |
| pc4 | 1458 | 37 | 7.19 | Binary | Software Engineering | NASA defect prediction data for flight software using code complexity metrics. |
| bank-marketing | 4521 | 48 | 7.68 | Binary | Finance | Direct marketing campaign data for predicting term deposit subscription. |
| Diabetes-130-Hospitals_(Fairlearn) | 101766 | 50 | 7.96 | Binary | Medicine | Hospital readmission prediction for diabetic patients based on 10 years of clinical records. |
| Otto-Group-Product-Classification-Challenge | 61878 | 93 | 8.36 | Multiclass | E-commerce | Multi-class product classification dataset from Otto Group with anonymized features. |
| eucalyptus | 4331 | 26 | 8.54 | Multiclass | Computer Systems | High-performance computing job scheduling dataset for predictive modeling. |
| pendigits | 10992 | 16 | 8.61 | Binary | Image Recognition | Binarized dataset for pen-based digit recognition. |
| pc3 | 1563 | 37 | 8.77 | Binary | Software Engineering | Defect prediction dataset from NASA flight software using complexity metrics. |
| page-blocks-bin | 5473 | 10 | 8.77 | Binary | Document Processing | Binarized version of page layout classification based on document blocks. |
| optdigits | 5620 | 64 | 8.83 | Binary | Image Recognition | Binarized optical digit recognition dataset from scanned documents. |
| mfeat-zernike | 2000 | 47 | 9.0 | Binary | Image Recognition | Zernike moments of handwritten digits, binarized for classification. |
| mfeat-fourier | 2000 | 76 | 9.0 | Binary | Image Recognition | Fourier coefficients of handwritten digits, binarized for classification. |
| mfeat-karhunen | 2000 | 64 | 9.0 | Binary | Image Recognition | Karhunen-Loeve coefficients of handwritten digits, binarized for classification. |
| Pulsar-Dataset-HTRU2 | 17898 | 8 | 9.92 | Binary | Astronomy | Binary classification of pulsar vs. non-pulsar signals from radio telescope data. |
| vowel | 990 | 26 | 10.0 | Binary | Speech Recognition | Binarized classification of vowel sounds based on audio features. |
| heart-h | 294 | 13 | 12.53 | Multiclass | Medicine | Hungarian heart disease data used to predict cardiac conditions. |
| pc1 | 1109 | 21 | 13.4 | Binary | Software Engineering | NASA flight software defect prediction dataset using McCabe and Halstead metrics. |
| seismic-bumps | 2584 | 22 | 14.2 | Binary | Geophysics | Predicts hazardous seismic events in coal mines based on geophysical monitoring data. |
| ozone-level-8hr | 2534 | 72 | 14.84 | Binary | Environmental Science | Forecasts peak ozone levels using meteorological and atmospheric features. |
| microaggregation2 | 20000 | 20 | 15.02 | Multiclass | Privacy Data Mining | Dataset used for evaluating microaggregation methods in privacy-preserving learning. |
| Sick_numeric | 3772 | 29 | 15.33 | Binary | Medicine | Numeric version of thyroid disease diagnosis data with binarized features. |
| insurance_company | 9822 | 85 | 15.76 | Binary | Finance | Predicts caravan insurance ownership using socio-demographic and product data. |
| wilt | 4839 | 5 | 17.54 | Binary | Remote Sensing | Remote sensing dataset for detecting diseased trees using multispectral imagery. |
| Click_prediction_small | 149639 | 11 | 21.37 | Binary | Advertising | Small-scale dataset for predicting ad click-throughs. |
| jannis | 83733 | 54 | 22.83 | Multiclass | Image Recognition | Classify image regions into one of the 4-most populated branches. |
| letter | 20000 | 16 | 23.6 | Binary | Image Recognition | Binarized handwritten letter recognition dataset. |
| walking-activity | 149332 | 4 | 24.14 | Multiclass | Biometrics | Accelerometer data used for user identification from walking patterns. |
| helena | 65196 | 27 | 36.08 | Multiclass | Image Recognition | Classify image regions into one of 100 labels. |
| mammography | 11183 | 6 | 42.01 | Binary | Medicine | Mammography dataset used for anomaly and breast cancer detection tasks. |
| dis | 3772 | 29 | 64.03 | Binary | Biology | Dataset from PMLB used for binary classification in biomedical domains. |
| Satellite | 5100 | 36 | 67.0 | Binary | Remote Sensing | Classifies land cover and detects anomalies in satellite image data. |
| Employee-Turnover-at-TECHCO | 34452 | 9 | 68.74 | Binary | Human Resources | Dataset modeling monthly employee turnover in a tech company. |
| page-blocks | 5473 | 10 | 175.46 | Multiclass | Document Processing | Page layout classification based on document blocks. |
| allbp | 3772 | 29 | 257.79 | Multiclass | Biology | Blood pressure data for classification, sourced from PMLB. |
| CreditCardFraudDetection | 284807 | 30 | 577.88 | Binary | Finance | Highly imbalanced dataset for detecting fraudulent credit card transactions. |

**Dataset source and access.** All datasets are hosted on the OpenML (Vanschoren et al., 2014) platform. We provide a wrapper function based on the OpenML API in the CLIMB Python package, allowing users to easily download the datasets and apply standardized preprocessing.

# B   More Reproducibility Details

**Hyperparameter search.** We used Optuna (Akiba et al., 2019) to search for the best configuration of CIL methods with tunable hyperparameters. Hyperparameter optimization was conducted for 23 out of 29 CIL methods on each dataset, with AUPRC as the optimization objective. Table 4 reports the hyperparameter search space for each method. Importantly, we observed that using a single random split to create a validation set often caused the selected hyperparameters to overfit, especially due to the scarcity of minority class samples. When conducting a more comprehensive 5-fold stratified evaluation, the hyperparameters found through such overfitted search frequently underperformed compared to the default settings. To address this issue, we adopted 5-fold stratified training and evaluation within each search trial, despite the increased computational cost. For each method-dataset pair, we performed 100 search iterations and employed an early stopping strategy with 10 rounds patience to improve efficiency. We use the Tree-structured Parzen Estimator (Ozaki et al., 2022) (`optuna.samplers.TPESampler`) to sample hyperparameters in each trail. Additionally, for every method and dataset, we also evaluated the performance using default hyperparameters. The final hyperparameters were chosen based on the better result between the search and the default setting. Running these hyperparameter searches consumed more than 500 hours on our workstation.

Table 4: Hyperparameter search spaces.

| Method | Search Parameters |
|---|---|
| NearMiss | `n_neighbors` $\in [1, 10]$ |
| EditedNearestNeighbors | `n_neighbors` $\in [1, 10]$, `kind_sel` $\in \{\text{all}, \text{mode}\}$ |
| Repeated ENN | `n_neighbors` $\in [1, 10]$, `kind_sel` $\in \{\text{all}, \text{mode}\}$ |
| AllKNN | `n_neighbors` $\in [1, 10]$, `kind_sel` $\in \{\text{all}, \text{mode}\}$ |
| OneSideSelection | `n_neighbors` $\in [1, 10]$ |
| NeighborhoodCleaningRule | `n_neighbors` $\in [1, 10]$, `kind_sel` $\in \{\text{all}, \text{mode}\}$, `threshold_cleaning` $\in [0.0, 1.0]$ |
| SMOTE | `k_neighbors` $\in [1, 10]$ |
| BorderlineSMOTE | `k_neighbors` $\in [1, 10]$, `m_neighbors` $\in [1, 10]$ |
| SVMSMOTE | `k_neighbors` $\in [1, 10]$, `m_neighbors` $\in [1, 10]$ |
| ADASYN | `n_neighbors` $\in [1, 10]$ |
| SelfPacedEnsemble | `k_bins` $\in [1, 10]$ |
| BalanceCascade | `replacement` $\in \{\text{True}, \text{False}\}$ |
| BalancedRandomForest | `max_samples` $\in [0.5, 1.0]$, `max_features` $\in [0.5, 1.0]$ |
| EasyEnsemble | `max_samples` $\in [0.5, 1.0]$, `max_features` $\in [0.5, 1.0]$ |
| RUSBoost | `learning_rate` $\in [0.0, 1.0]$, `algorithm` $\in \{\text{SAMME}, \text{SAMME.R}\}$ |
| UnderBagging | `max_samples` $\in [0.5, 1.0]$, `max_features` $\in [0.5, 1.0]$ |
| OverBoost | `learning_rate` $\in [0.0, 1.0]$, `algorithm` $\in \{\text{SAMME}, \text{SAMME.R}\}$ |
| OverBagging | `max_samples` $\in [0.5, 1.0]$, `max_features` $\in [0.5, 1.0]$ |
| SMOTEBoost | `learning_rate` $\in [0.0, 1.0]$, `algorithm` $\in \{\text{SAMME}, \text{SAMME.R}\}$, `k_neighbors` $\in [1, 10]$ |
| SMOTEBagging | `max_samples` $\in [0.5, 1.0]$, `max_features` $\in [0.5, 1.0]$, `k_neighbors` $\in [1, 10]$ |
| AdaCost | `learning_rate` $\in [0.0, 1.0]$, `algorithm` $\in \{\text{SAMME}, \text{SAMME.R}\}$ |
| AdaUBoost | `learning_rate` $\in [0.0, 1.0]$, `algorithm` $\in \{\text{SAMME}, \text{SAMME.R}\}$ |
| AsymBoost | `learning_rate` $\in [0.0, 1.0]$, `algorithm` $\in \{\text{SAMME}, \text{SAMME.R}\}$ |

**Dataset preprocessing and split.** As described in Section 3.3, to mitigate the randomness introduced by a single random train-test split, we adopt a 5-fold stratified splitting strategy for all datasets and report the average performance. We use the `sklearn.model_selection.StratifiedKFold` utility from scikit-learn (Pedregosa et al., 2011) to obtain stratified folds that preserve the percentage of samples in each class, ensuring that the imbalance ratio remains consistent across all splits. Similarly, we apply `preprocessing.StandardScaler` to standardize numerical features. For categorical features, we use `OrdinalEncoder` for binary attributes and `OneHotEncoder` for multi-class attributes.

**Runtime measurement.** The runtime reported in Figure 3 was measured on a Windows workstation equipped with an Intel Core i9-12900 CPU. It reflects the training time for a **single split** in a 5-fold stratified split, that is, the training data is formed by 4 out of 5 splits (80%). Therefore, the total runtime for each hyperparameter search should be further multiplied by 5 splits and 100 iterations.

**Performance-runtime analysis with all metrics.** Similarly, due to space constraints, we only visualized the performance-runtime trade-off based on AUPRC in the main paper (Figure 3). Here,

we provide additional visualizations based on F1-score (Figure 6) and balanced accuracy (Figure 7). While minor changes in the ranking of some methods can be observed, the differences across method branches remain significant. Thus, the related analyses and Takeaway #4 in the main text still hold: undersample ensembles continue to represent the most effective category for achieving the best performance-efficiency trade-off.

## C  Additional Experiments, Detailed Results, and Discussions

### C.1  Results with Additional CIL methods

Table 5:  Extended summary benchmark results with hybrid sampling methods (SMOTEENN, SMOTETomek) and GBDTs (XGBoost, LightGBM, CATBoost), this table extends the main results in Table 2 by including additional CIL baseline. Given the large number of results, we group the 73 datasets by imbalance level into 4 categories and report the averaged AUPRC (AP), macro F1, and Balanced Accuracy (BAC) for each CIL method (in $\times 10^{-2}$). Detailed results for each dataset can be found in C. For a comprehensive evaluation, we also rank all methods on each dataset and metric, and report their average ranks. **Color coding is used to show the performance gains (blue) or losses (red) relative to the base no-balancing method, with deeper colors indicating larger differences.**

| Dataset Group | Avg. Stat | Metric | Base | RUS | CC | IHT | NM | TL | ENN | RENN | AKNN | OSS | NCR | ROS | SMT | BSMT | SSMT | ASYN | SENN | STom | SPE | BC | BRF | EE | UBS | UBA | OBS | SMBS | OBA | SMBA | AdaC | AdaBS | AsyBS | CS | XGB | LGB | CAT |
|---|---|---|---|---|---|---|---|---|---|---|---|---|---|---|---|---|---|---|---|---|---|---|---|---|---|---|---|---|---|---|---|---|---|---|---|---|---|
| IR ∈ [0, 5) (28 datasets) | Score (↑) | AP | 51.0 | 49.4 | 48.0 | 45.8 | 45.5 | 51.5 | 53.6 | 53.4 | 52.9 | 51.7 | 53.7 | 51.1 | 51.6 | 51.8 | 52.1 | 51.6 | 51.8 | 51.2 | 59.3 | 57.8 | 57.7 | 59.0 | 58.6 | 59.0 | 52.9 | 53.7 | 58.4 | 58.9 | 52.5 | 52.4 | 52.5 | 51.2 | 58.0 | 58.2 | 58.3 |
| | | F1 | 72.0 | 70.4 | 67.1 | 66.0 | 65.5 | 72.8 | 73.9 | 73.3 | 73.0 | 73.0 | 74.2 | 72.4 | 72.9 | 72.9 | 73.2 | 72.8 | 72.6 | 72.5 | 77.9 | 76.8 | 76.7 | 78.0 | 77.5 | 78.0 | 73.7 | 74.3 | 76.5 | 77.4 | 73.2 | 73.2 | 73.2 | 72.3 | 76.4 | 76.3 | 75.7 |
| | | BAC | 72.1 | 73.3 | 71.7 | 73.8 | 69.7 | 73.3 | 76.5 | 76.7 | 76.4 | 73.6 | 76.6 | 72.5 | 73.5 | 73.6 | 73.8 | 73.6 | 75.5 | 73.1 | 78.5 | 77.3 | 79.7 | 79.7 | 78.7 | 79.7 | 73.7 | 74.8 | 75.7 | 77.0 | 73.2 | 73.1 | 73.2 | 72.4 | 75.8 | 75.9 | 75.5 |
| | Rank (↓) | AP | 24.8 | 29.3 | 31.4 | 27.1 | 30.6 | 22.5 | 15.4 | 16.4 | 18.5 | 21.6 | 15.5 | 25.2 | 23.4 | 22.8 | 22.5 | 23.8 | 21.8 | 24.7 | 6.0 | 10.0 | 9.6 | 6.8 | 6.9 | 6.6 | 19.6 | 17.0 | 8.0 | 6.1 | 20.0 | 20.9 | 20.2 | 25.7 | 10.1 | 9.6 | 9.6 |
| | | F1 | 24.1 | 30.4 | 32.0 | 33.0 | 32.0 | 21.2 | 16.4 | 19.1 | 21.1 | 19.9 | 16.0 | 24.2 | 22.5 | 21.8 | 20.6 | 22.3 | 22.2 | 23.3 | 5.0 | 8.9 | 10.2 | 5.7 | 6.7 | 5.6 | 19.0 | 15.7 | 10.2 | 6.6 | 19.1 | 20.8 | 19.3 | 24.9 | 9.9 | 9.4 | 10.9 |
| | | BAC | 28.3 | 24.0 | 28.1 | 12.7 | 29.5 | 22.8 | 13.2 | 12.7 | 14.2 | 21.4 | 13.1 | 27.5 | 23.5 | 22.5 | 20.9 | 22.4 | 17.1 | 24.2 | 7.1 | 9.9 | 3.9 | 3.9 | 6.2 | 3.7 | 23.0 | 17.2 | 14.4 | 10.1 | 22.9 | 25.0 | 23.5 | 28.7 | 14.0 | 13.5 | 14.9 |
| IR ∈ [5, 10) (24 datasets) | Score (↑) | AP | 50.9 | 43.2 | 35.3 | 40.0 | 32.1 | 51.0 | 52.5 | 52.6 | 52.5 | 50.8 | 52.5 | 51.1 | 51.7 | 51.4 | 51.9 | 50.7 | 50.8 | 50.7 | 64.6 | 62.7 | 60.5 | 62.4 | 63.8 | 62.4 | 54.8 | 54.1 | 61.2 | 62.8 | 54.1 | 54.4 | 54.1 | 51.4 | 62.8 | 63.2 | 62.5 |
| | | F1 | 74.7 | 68.4 | 56.7 | 64.2 | 57.4 | 74.6 | 75.1 | 75.0 | 75.0 | 74.5 | 75.2 | 74.4 | 75.1 | 75.0 | 75.3 | 74.6 | 73.6 | 74.6 | 79.7 | 78.7 | 76.7 | 78.7 | 79.3 | 78.7 | 75.6 | 75.8 | 77.0 | 78.5 | 75.4 | 75.4 | 75.4 | 74.5 | 78.2 | 78.3 | 77.3 |
| | | BAC | 74.7 | 76.0 | 71.2 | 76.5 | 70.8 | 74.9 | 77.6 | 78.0 | 77.7 | 74.8 | 77.8 | 74.4 | 76.4 | 75.8 | 76.4 | 75.7 | 77.8 | 76.0 | 82.1 | 80.8 | 83.1 | 82.9 | 82.5 | 82.9 | 75.4 | 76.8 | 75.6 | 77.7 | 75.0 | 75.1 | 75.0 | 74.5 | 77.0 | 77.0 | 76.0 |
| | Rank (↓) | AP | 24.4 | 29.9 | 33.3 | 29.5 | 33.3 | 23.8 | 17.8 | 17.2 | 18.4 | 23.8 | 17.8 | 23.9 | 21.3 | 22.3 | 20.5 | 23.9 | 23.0 | 23.6 | 3.5 | 7.5 | 10.6 | 7.8 | 4.3 | 7.8 | 17.2 | 18.2 | 12.1 | 8.0 | 18.4 | 18.6 | 18.8 | 23.6 | 8.9 | 7.6 | 9.6 |
| | | F1 | 22.5 | 32.0 | 34.2 | 33.3 | 34.2 | 21.9 | 18.7 | 18.6 | 18.6 | 22.2 | 18.3 | 22.2 | 19.7 | 21.0 | 18.3 | 22.4 | 24.7 | 21.9 | 4.5 | 7.9 | 15.4 | 8.1 | 4.5 | 7.8 | 16.9 | 16.8 | 13.3 | 7.2 | 17.4 | 18.1 | 17.8 | 21.9 | 8.7 | 8.6 | 10.5 |
| | | BAC | 26.3 | 21.6 | 29.3 | 20.0 | 29.5 | 25.2 | 16.4 | 15.4 | 16.1 | 24.5 | 16.1 | 26.5 | 17.7 | 21.5 | 17.6 | 22.0 | 14.9 | 19.4 | 5.9 | 7.5 | 3.2 | 3.9 | 3.7 | 3.2 | 22.1 | 16.7 | 20.6 | 13.5 | 24.0 | 24.2 | 24.3 | 26.1 | 16.5 | 16.4 | 18.2 |
| IR ∈ [10, 50) (15 datasets) | Score (↑) | AP | 34.9 | 23.6 | 17.4 | 27.1 | 14.5 | 35.1 | 36.2 | 36.2 | 35.9 | 35.5 | 36.3 | 34.1 | 34.3 | 35.8 | 35.0 | 34.1 | 33.2 | 33.7 | 47.1 | 41.4 | 38.4 | 41.9 | 44.9 | 41.9 | 36.7 | 36.6 | 45.0 | 46.0 | 36.4 | 36.7 | 36.4 | 34.1 | 47.3 | 43.8 | 45.1 |
| | | F1 | 61.6 | 51.2 | 35.8 | 52.9 | 35.6 | 61.6 | 62.3 | 61.9 | 61.5 | 61.9 | 62.4 | 61.0 | 61.0 | 62.1 | 61.7 | 61.0 | 59.5 | 60.3 | 65.5 | 62.1 | 59.7 | 62.9 | 64.0 | 62.9 | 61.6 | 61.7 | 64.0 | 65.8 | 61.5 | 61.6 | 61.5 | 61.0 | 65.7 | 63.0 | 63.6 |
| | | BAC | 61.8 | 63.6 | 56.7 | 65.8 | 54.3 | 62.1 | 64.5 | 65.0 | 64.3 | 62.3 | 64.1 | 60.8 | 63.5 | 63.7 | 63.5 | 63.2 | 65.2 | 63.0 | 70.7 | 67.8 | 72.2 | 72.1 | 70.4 | 72.1 | 61.1 | 64.0 | 62.4 | 65.1 | 61.5 | 61.2 | 61.5 | 61.0 | 64.1 | 61.6 | 62.1 |
| | Rank (↓) | AP | 22.0 | 29.9 | 34.3 | 25.3 | 34.2 | 21.4 | 16.0 | 16.3 | 17.3 | 19.9 | 15.3 | 24.7 | 23.3 | 18.5 | 20.5 | 23.3 | 22.2 | 24.7 | 4.7 | 11.1 | 16.0 | 11.7 | 6.8 | 11.6 | 17.1 | 17.5 | 9.7 | 7.4 | 19.5 | 17.6 | 20.1 | 24.5 | 5.0 | 10.4 | 10.0 |
| | | F1 | 17.0 | 32.8 | 34.4 | 31.7 | 34.2 | 17.7 | 12.7 | 14.5 | 16.9 | 15.7 | 13.7 | 21.6 | 21.4 | 14.5 | 17.6 | 21.3 | 25.3 | 23.3 | 5.7 | 16.7 | 22.7 | 14.0 | 9.0 | 13.7 | 17.2 | 17.9 | 12.1 | 6.5 | 18.1 | 17.5 | 18.1 | 21.3 | 6.7 | 13.7 | 13.1 |
| | | BAC | 23.1 | 19.2 | 31.6 | 13.7 | 32.3 | 22.5 | 16.7 | 15.1 | 17.5 | 21.7 | 16.7 | 27.7 | 17.0 | 15.3 | 17.1 | 19.9 | 13.9 | 18.7 | 5.6 | 7.8 | 2.9 | 2.8 | 5.7 | 2.3 | 26.4 | 16.3 | 20.7 | 15.5 | 24.1 | 26.3 | 24.9 | 26.7 | 15.9 | 23.4 | 22.6 |
| IR ∈ [50, 1000) (6 datasets) | Score (↑) | AP | 42.6 | 18.9 | 15.9 | 33.0 | 13.5 | 44.2 | 45.0 | 44.1 | 44.9 | 44.3 | 44.6 | 41.7 | 37.1 | 42.2 | 41.9 | 34.3 | 34.7 | 34.4 | 57.5 | 50.1 | 32.9 | 35.5 | 43.3 | 35.5 | 45.0 | 40.5 | 56.4 | 56.0 | 48.1 | 46.6 | 48.1 | 41.7 | 58.9 | 48.9 | 58.1 |
| | | F1 | 74.0 | 50.6 | 35.2 | 68.0 | 32.9 | 75.0 | 75.1 | 74.7 | 74.8 | 75.1 | 75.0 | 73.6 | 71.9 | 74.4 | 73.7 | 70.3 | 70.2 | 70.1 | 74.7 | 68.6 | 56.3 | 59.8 | 68.5 | 59.8 | 74.3 | 71.9 | 75.8 | 76.6 | 74.9 | 74.9 | 74.9 | 73.9 | 78.7 | 73.4 | 77.8 |
| | | BAC | 74.6 | 81.8 | 70.5 | 79.9 | 66.2 | 75.1 | 75.8 | 76.1 | 75.6 | 75.2 | 75.6 | 73.1 | 77.2 | 76.1 | 76.0 | 76.3 | 79.4 | 76.0 | 85.9 | 83.0 | 88.0 | 87.3 | 85.7 | 87.3 | 73.5 | 77.6 | 72.4 | 74.4 | 74.8 | 73.2 | 74.8 | 73.1 | 75.8 | 73.5 | 74.9 |
| | Rank (↓) | AP | 19.2 | 32.7 | 34.2 | 24.0 | 34.8 | 19.5 | 14.8 | 16.0 | 15.7 | 16.7 | 17.8 | 22.0 | 21.0 | 17.8 | 19.5 | 25.2 | 26.2 | 24.3 | 4.5 | 10.8 | 21.0 | 18.8 | 12.7 | 18.5 | 16.7 | 19.3 | 9.8 | 9.0 | 13.0 | 16.2 | 13.3 | 21.3 | 6.7 | 10.2 | 6.8 |
| | | F1 | 16.0 | 32.8 | 34.5 | 22.5 | 34.5 | 14.0 | 11.5 | 14.2 | 13.7 | 12.2 | 13.8 | 18.2 | 17.5 | 14.5 | 16.5 | 22.0 | 21.0 | 20.8 | 13.3 | 19.3 | 31.8 | 30.2 | 22.7 | 30.2 | 15.3 | 16.5 | 12.8 | 10.7 | 11.8 | 14.0 | 11.5 | 15.7 | 5.5 | 10.7 | 7.8 |
| | | BAC | 24.3 | 7.8 | 24.8 | 11.3 | 26.5 | 23.7 | 15.7 | 16.5 | 16.7 | 22.3 | 15.8 | 29.3 | 16.2 | 18.5 | 18.5 | 19.5 | 10.7 | 17.3 | 5.5 | 5.8 | 2.3 | 0.3 | 4.3 | 2.7 | 22.3 | 15.8 | 22.7 | 20.5 | 22.3 | 28.7 | 24.0 | 28.5 | 19.8 | 24.7 | 23.5 |

*Abbreviations: Random Undersampling (**RUS**), Cluster Centroids (**CC**), Instance Hardness Threshold (**IHT**), NearMiss (**NM**), Tomek Links (**TL**), Edited Nearest Neighbors (**ENN**), Repeated ENN (**RENN**), AllKNN (**AKNN**), One-Sided Selection (**OSS**), Neighborhood Cleaning Rule (**NCR**), Random Oversampling (**ROS**), SMOTE (**SMT**), Borderline SMOTE (**BSMT**), SVM SMOTE (**SSMT**), ADASYN (**ASYN**), SMOTEENN (**SENN**), SMOTE Tomek (**STom**), Self-paced Ensemble (**SPE**), Balance Cascade (**BC**), Balanced Random Forest (**BRF**), Easy Ensemble (**EE**), RUSBoost (**UBS**), UnderBagging (**UBA**), OverBoost (**OBS**), SMOTEBoost (**SMBS**), OverBagging (**OBA**), SMOTEBagging (**SMBA**), Cost-sensitive (**CS**), AdaCost (**AdaC**), AdaUBoost (**AdaBS**), AsymBoost (**AsyBS**), XGBoost (**XGB**), LightGBM (**LGB**), CATBoost (**CAT**).

Here, we provide additional benchmark results that incorporate hybrid sampling methods (SMO-TEENN (Batista et al., 2004), SMOTETomek (Batista et al., 2003)) and popular gradient-boosted decision tree (GBDT) models (XGBoost (Chen and Guestrin, 2016), LightGBM (Ke et al., 2017), CatBoost (Hancock and Khoshgoftaar, 2020)). These methods were not discussed in detail in the main paper, as our primary focus is on establishing a fair and unified comparison of representative class-imbalanced learning (CIL) techniques under consistent experimental settings. Hybrid sampling methods tend to be slower, more complex, and often underperform relative to their simpler counterparts, while GBDT models rely on specialized base learners with optimization strategies that differ fundamentally from the scikit-learn trees used throughout our benchmark, making direct comparisons less meaningful. But still, we include these results here for completeness, as they were frequently raised during the review process and help further contextualize the scope and applicability of CLIMB.

For implementation, SMOTEENN and SMOTETomek are adopted from the `imblearn` (Lemaître et al., 2017) package with their default configurations, ensuring consistency with widely used practice. For GBDTs, we evaluate XGBoost, LightGBM, and CatBoost under the same ensemble size as

the other ensemble-based CIL methods in the main paper. The extended results are summarized in Table 5. The latter full dataset-level results in Table 6-9 also include these new CIL methods.

**Hybrid Sampling Methods.** The extended results highlight two consistent trends. First, the hybrid sampling methods SMOTEENN and SMOTETomek do not provide consistent benefits across imbalance levels. Their performance in terms of AP, F1, and BAC is typically comparable to or worse than their single-component counterparts (e.g., SMOTE or ENN/Tomek alone), and their average ranks remain relatively low. This confirms that the added complexity of combining oversampling and cleaning does not yield robust gains in practice.

**Advanced GBDTs.** Second, the GBDT baselines (XGBoost, LightGBM, CatBoost) achieve strong overall results, often surpassing classical resampling-based methods, particularly under higher imbalance ratios. Nevertheless, they are not uniformly superior: ensemble-based CIL methods such as SPE, RUSBoost, and SMOTEBagging remain highly competitive, achieving comparable or better ranks in several imbalance groups. These findings indicate that while GBDTs constitute powerful baselines, well-designed CIL ensembles can match or exceed their performance, especially when tailored to severe imbalance scenarios.

## C.2 Pairwise comparisons between all CIL methods.

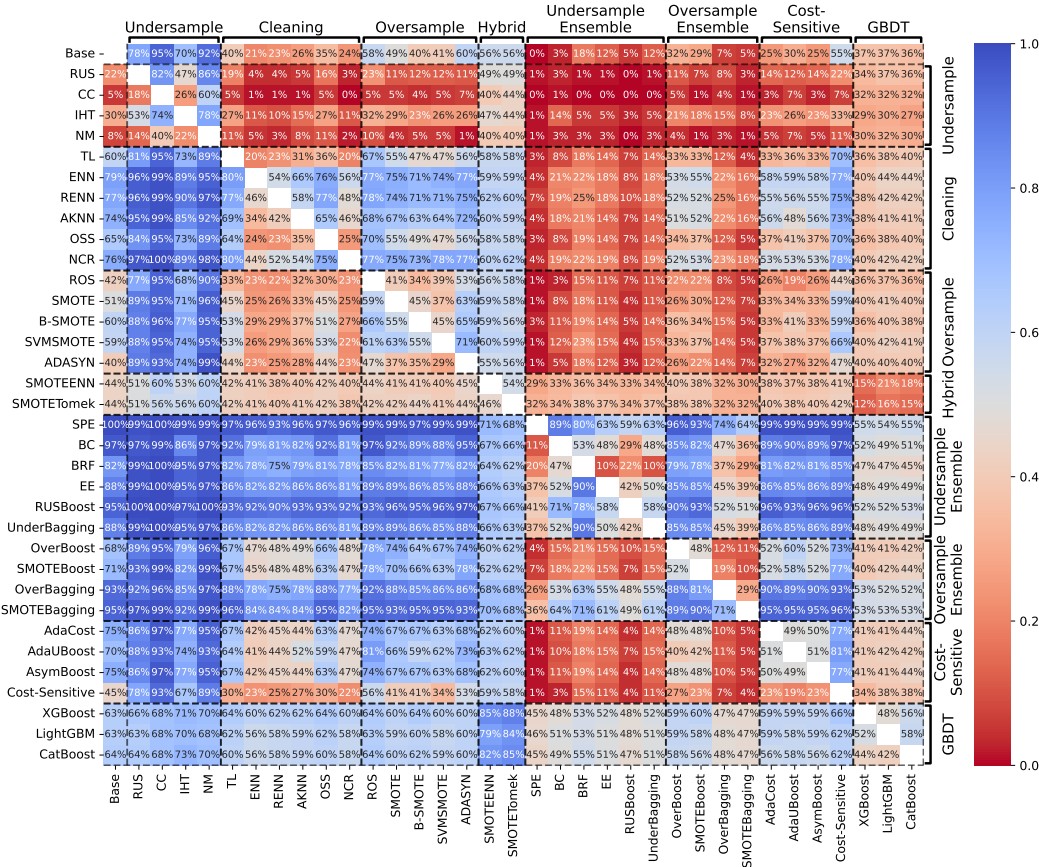

Figure 5: **Pair-wise win ratio** (by AUPRC) comparison between all CIL algorithms. The number represents the ratio of datasets that the **row method outperforms the column method** on, i.e., a blue/red row means the row method consistently outperforms/underperforms others.

To provide more detailed insights for model selection, we pair each combination of two CIL methods (denoted as A and B) and compute the proportion of datasets where method A outperforms method B. The results based on AUPRC are shown in Figure 5. Consistent with the analysis in Section 4.1, ensemble methods generally demonstrate a consistent advantage over non-ensemble methods across

most datasets. Among the ensemble approaches, SPE, OverBagging, and SMOTEBagging achieve relatively high win ratios. In particular, SPE, an efficient undersampling-based ensemble method, maintains a win ratio above 59% against all other CIL methods. This highlights the potential of ensemble approaches that incorporate informed undersampling strategies.

## C.3 Discussion on the Self-paced Ensemble

Among all the evaluated CIL methods, Self-paced Ensemble (SPE) stands out as the most consistent top performer. Its advantage can be attributed to a few complementary factors.

- **Hard example mining:** SPE retains difficult-to-classify samples during undersampling, which improves decision boundaries.
- **Noise robustness:** The hardness harmonization mechanism balances informativeness and noise, avoiding the inclusion of overly noisy samples.
- **Self-paced learning:** Inspired by curriculum learning, SPE introduces samples progressively from easy to hard, which stabilizes training.
- **Efficiency:** As an undersampling-based ensemble, SPE trains on fewer samples per model, making it more efficient than oversampling or boosting strategies.

## C.4 Comparison with BAF Benchmark

We also compare CLIMB with the BAF benchmark (Jesus et al., 2022). Both address imbalance, but with different goals and setups. CLIMB evaluates CIL methods on 73 real-world datasets with natural imbalance, using repeated cross-validation and standard metrics such as AUPRC, Macro-F1, and Balanced Accuracy. BAF instead focuses on fairness under distributional bias and temporal shift in a single fraud detection task, using CTGAN-generated synthetic data, temporal splits, and fairness metrics like TPR@5%FPR and FPR ratio. Thus, CLIMB offers a broad benchmark for CIL effectiveness, while BAF targets fairness in a specific application.

## C.5 Detailed main results on each dataset.

Due to space constraints, in the main results (Table 2), we reported the average scores and rankings for each metric by grouping the 73 datasets into four categories based on their imbalance levels. Here, we provide the complete results for each method on each individual dataset. Specifically, AUPRC, F1-score, and Balanced Accuracy results are reported in Tables 6, 7, and 8, respectively. Additionally, Table 9 presents the runtime of each method across different datasets. The dataset ordering in these tables follows the order defined in Table 3. We used color coding similar to Table 2 (i.e., blue represents better than no balancing, and red represents worse than no balancing, with deeper colors indicating larger differences) for improved clarity.

**Dataset-level Analysis.** Although the overall conclusions of CLIMB are robust across datasets, a few cases deviate from the general trends. We intentionally phrased our main takeaways to avoid overgeneralization, and here we highlight notable examples to provide additional context:

- **Undersampling ensembles on extremely imbalanced datasets (e.g., *dis*, *satellite*):** Random undersampling based ensembles such as Balanced Random Forest (BRF), EasyEnsemble, and UBS can fail when the imbalance ratio is very severe. These methods discard most majority class samples, which results in insufficient training information and weak generalization. In contrast, approaches like Self-paced Ensemble (SPE) and BalanceCascade (BC) are more robust because they explicitly retain informative samples through hard example mining.
- **Cleaning based methods on long-tailed multiclass datasets (e.g., *user-knowledge*, *allbp*):** Cleaning based methods such as Tomek Links, ENN, and RENN often underperform in long-tailed multiclass scenarios. Since multiple minority classes can be close to majority classes in feature space, these cleaning procedures tend to over remove minority samples. This reduces the model's ability to learn rare class patterns and leads to degraded performance.

These exceptions are limited in scope and do not alter the overall conclusions of our study. Instead, they illustrate the importance of understanding dataset specific characteristics when selecting and applying CIL methods in practice.

Table 6: Detailed full results on each dataset on AUPRC ($\times 10^{-2}$).

| Dataset | Base | Undersample | | | | Cleaning | | | | | | Oversample | | | | | Hybrid | | Undersample Ensemble | | | | | | Oversample Ensemble | | | | Cost-Sensitive | | GBDTs | | | |
|---|---|---|---|---|---|---|---|---|---|---|---|---|---|---|---|---|---|---|---|---|---|---|---|---|---|---|---|---|---|---|---|---|---|---|
| | | RUS | CC | IHT | NM | TL | ENN | RENN | AKNN | OSS | NCR | ROS | SMT | BSMT | SSMT | ASYN | SENN | STom | SPE | BC | BRF | EE | UBS | UBA | OBS | SMBS | OBA | SMBA | AdaC | AdaBS | AsyBS | CS | XGB | LGB | CAT |

Table 7: Detailed full results on each dataset on macro F1 ($\times 10^{-2}$).

| Dataset | Base | Undersample | | | | Cleaning | | | | | | Oversample | | | | | Hybrid | | Undersample Ensemble | | | | | | Oversample Ensemble | | | | Cost-Sensitive | | GBDTs | | | |
|---|---|---|---|---|---|---|---|---|---|---|---|---|---|---|---|---|---|---|---|---|---|---|---|---|---|---|---|---|---|---|---|---|---|---|
| | | RUS | CC | IHT | NM | TL | ENN | RENN | AKNN | OSS | NCR | ROS | SMT | BSMT | SSMT | ASYN | SENN | STom | SPE | BC | BRF | EE | UBS | UBA | OBS | SMBS | OBA | SMBA | AdaC | AdaBS | AsyBS | CS | XGB | LGB | CAT |

Table 8: Detailed full results on each dataset on Balanced Accuracy ($\times 10^{-2}$).

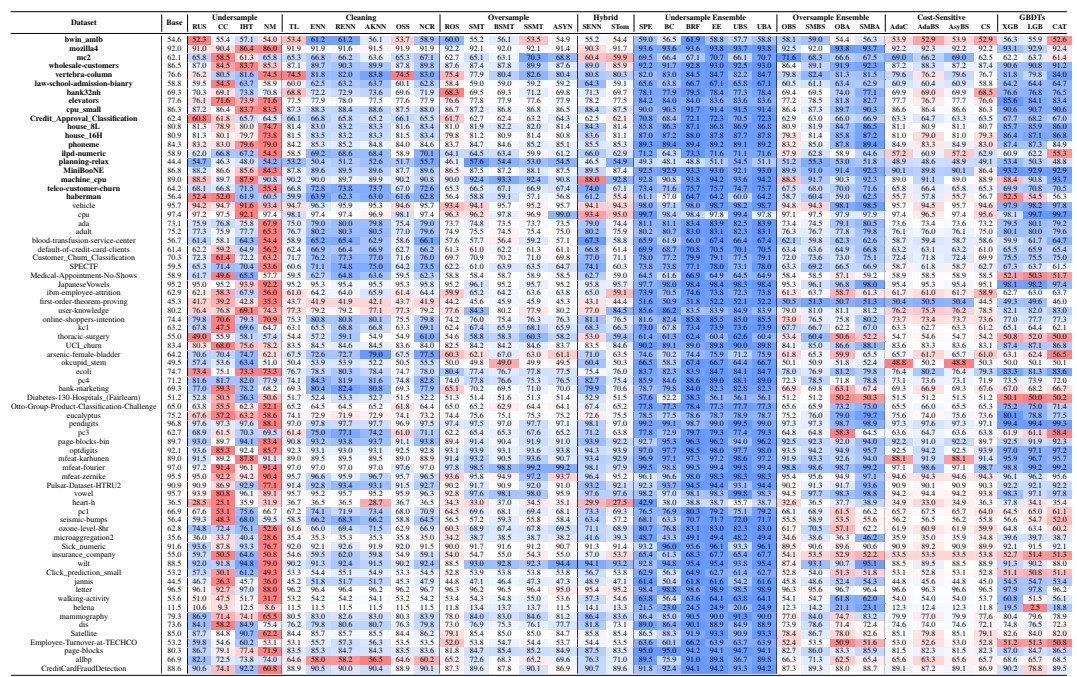

Table 9: Detailed full results on each dataset on runtime (ms).

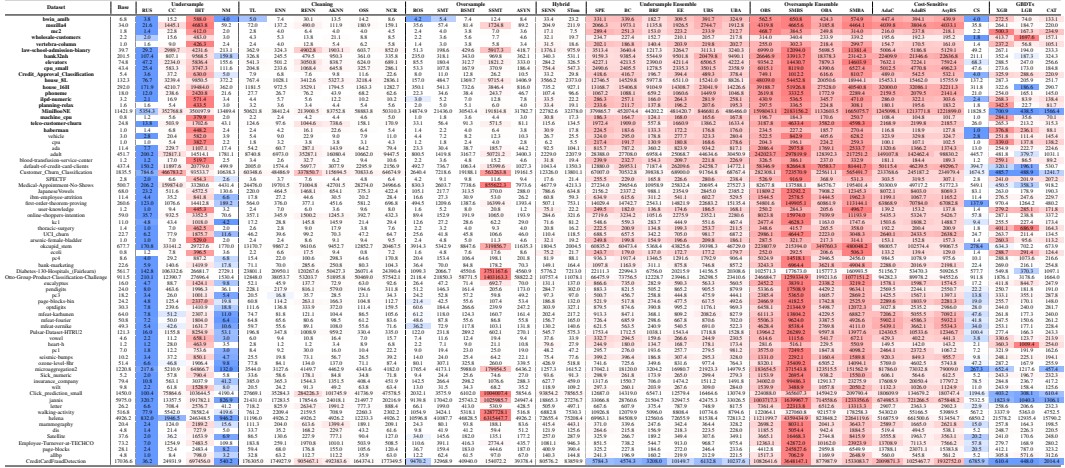

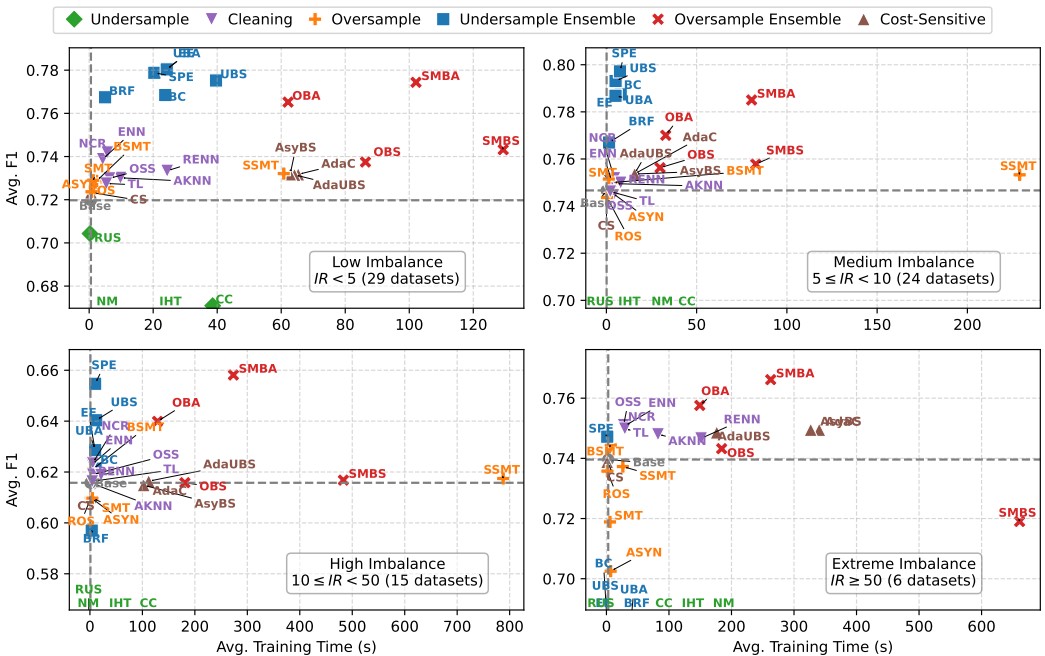

Figure 6: Macro F1 score versus runtime analysis, following the dataset grouping in Table 2. The **x-axis** shows the average runtime of each CIL algorithm, and the **y-axis** shows the average AUPRC score. **Desired methods are closer to the upper-left corner with high performance and low cost.** Different markers indicate different CIL method categories, the gray dashed line denotes the base model (no balancing) performance and runtime.

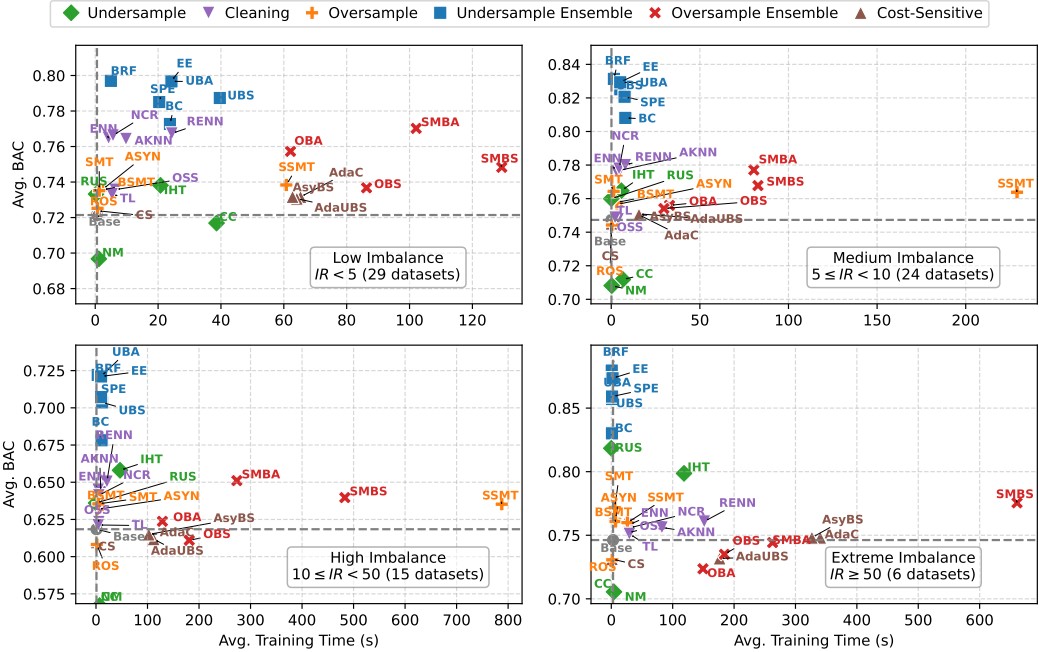

Figure 7: Balanced Accuracy versus runtime analysis, following the dataset grouping in Table 2. The **x-axis** shows the average runtime of each CIL algorithm, and the **y-axis** shows the average AUPRC score. **Desired methods are closer to the upper-left corner with high performance and low cost.** Different markers indicate different CIL method categories, the gray dashed line denotes the base model (no balancing) performance and runtime.

