# OpenReview forum: "CLIMB: Class-imbalanced Learning Benchmark on Tabular Data"
_NeurIPS.cc/2025/Datasets_and_Benchmarks_Track — NeurIPS 2025 Datasets and Benchmarks Track poster_

### Official Review · Reviewer_G91d · 2025-06-24

**Rating:** 4
**Confidence:** 4

**Summary:**

This paper introduces CLIMB, a comprehensive benchmark designed to address the challenges of class-imbalanced learning (CIL) on tabular data. CLIMB consists of 73 real-world datasets spanning various domains with different imbalance levels and 29 representative CIL algorithms, including resampling, cost-sensitive learning, and ensemble-based methods. The benchmark provides a unified API, high-quality documentation, and extensive support for easy implementation and comparison of algorithms. Through empirical analysis, the paper offers valuable insights into the effectiveness of different methods

**Dataset Code Accessibility:**

Yes

**Dataset Code Comments:**

The dataset and code are publicly available.

**Ethical Considerations:**

No, there are no or only very minor ethics concerns

**Final Justification:**

The response is reasonable. Parts of my concern are resolved. I raise the rating to borderline accept.

**Limitations Weaknesses:**

1. The paper would benefit from incorporating more advanced techniques for addressing imbalanced learning specific to tabular data. The insights presented are somewhat similar to those observed in the field of computer vision with regard to imbalanced learning. In vision-based tasks, it has been clearly demonstrated that traditional methods for handling imbalanced learning often lead to a significant performance drop on certain metrics.

2. Takeaway #3 does not seem to offer a novel conclusion or observation specific to this paper. The idea that different evaluation metrics highlight distinct aspects of performance is well-established in the field. As a result, this takeaway does not provide new insights and appears to be more of a general principle rather than a unique finding of the paper.

3. In Takeaway #5, the rationale for combining the two issues—noisy data and class imbalance—into a single experiment is not entirely clear. These issues are typically treated as separate problems, each with its own set of specific techniques.

4. The open-source library mentioned in the paper was previously introduced in the work by Imbens: Ensemble Class-imbalanced Learning in Python. Therefore, it is questionable whether it can be considered an original contribution of this study.

In conclusion, while the paper provides a comprehensive benchmark of various datasets and techniques for addressing imbalanced learning in tabular data, it lacks novel and insightful conclusions related to this problem.

**Strengths Contributions:**

The motivation behind the paper is highly compelling, addressing a critical gap in imbalanced learning for tabular data.

The writing is clear, well-organized, and easy to follow.

The experiments are thorough and ensure reproducibility.

---

> ### Author Rebuttal · Authors · 2025-07-29
>
> **Thank you for the careful review and helpful feedback! We've conducted new experiments to address your concerns and updated the paper accordingly. However, please understand that due to this year’s no‑link policy, we are unable to include new figures or links to new results/updated pdf. We try our best to summarize the key insights in text.**
>
> **Below we provide the point‑by‑point responses. For your convenience, we provied TL;DR summaries before lengthy detailed explanations to help you quickly grasp our main points.**
>
> # Q1.1: Incorporating more advanced techniques
>
> > The paper would benefit from incorporating more advanced techniques for addressing imbalanced learning specific to tabular data.
>
> ## TL;DR
>
> **We show that the CIL methods in CLIMB outperform modern GBDT models that are still SOTA for tabular data learning tasks. We also welcome suggestions of specific newer methods for inclusion.**
>
> ## Response
>
> In preparing this benchmark, we carefully reviewed both classical and recent literature but found that most recent techniques are tailored for deep models or designed for non-tabular modalities (e.g., images, text), making them incompatible with the scope and design goals of CLIMB.
>
> Nonetheless, recent studies [1,2] have reaffirmed the dominance of tree-based models (e.g., XGBoost, LightGBM) for tabular data. Motivated by this, we conducted additional experiments to validate our CIL methods against/with these strong GBDT baselines. We updated Figure 5 (model-to-model win rate) to include XGBoost, LightGBM, and CatBoost using the same ensemble size as other methods. We observe that several of our evaluated CIL methods (e.g., SPE, RUSBoost, and SMOTEBagging) consistently outperform these GBDTs in AUPRC and macro-F1, even with less-optimized scikit-learn trees.
>
> |WinRate vs.|XGB-AP|XGB-F1|XGB-BACC|LGB-AP|LGB-F1|LGB-BACC|CAT-AP|CAT-F1|CAT-BACC|
> |---|---|---|---|---|---|---|---|---|---|
> |SPE|65.8%|67.1%|89.0%|58.2%|63.7%|84.2%|61.0%|67.8%|89.7%|
> |UBS|54.8%|52.1%|93.2%|53.4%|54.8%|93.2%|54.8%|58.9%|94.5%|
> |SMBA|45.2%|52.1%|57.5%|45.9%|57.5%|66.4%|50.0%|59.6%|73.3%|
>
> **That said, we appreciate the reviewer’s suggestion. If there are specific recent CIL methods you believe we should compare against, we would be happy to include them in our discussion.**
>
> # Q1.2: Distinctions between CV and tabular domain
>
> > “The insights presented are somewhat similar to those observed in the field of computer vision with regard to imbalanced learning. In vision-based tasks, it has been clearly demonstrated that traditional methods for handling imbalanced learning often lead to a significant performance drop on certain metrics.”
>
> ## TL;DR
> **While some high-level CIL insights are shared with the vision domain, many of our findings are specific to the tabular setting, where traditional methods like resampling, cleaning, and tree-based ensembles remain effective. Differences in model architectures, data regimes, and practical constraints lead to distinct behaviors and insights not commonly addressed in vision-oriented studies.**
>
> ## Response
>
> **We believe that the observation you raised "traditional CIL methods often lead to significant performance drops in vision tasks" actually highlights one of a key difference between the two domains**: on tabular data, many of these "traditional" CIL methods (e.g., cleaning and oversampling techniques) are still effective and can lead to performance gains. There are also several important distinctions between tabular and vision domains:
> 1. In modern vision tasks, the dominant models are deep architectures such as CNNs and Vision Transformers. Accordingly, most CIL techniques developed for vision are designed specifically for training neural networks. These include loss function engineering to emphasize minority classes (e.g., Focal Loss [1], Class-Balanced Loss [2]), or network designs that explicitly decouple learning for head and tail classes [3, 4].
> 2. In contrast, tree-based models remain the most widely used approach for tabular data. This choice is supported by empirical studies [5] showing that tree-based methods still outperform deep learning on most tabular datasets due to their favorable inductive biases.
>
> **Because of this architectural divergence between the two domains, many findings in our work are less discussed in the vision literature. For example:**
> - **The role of large-scale model ensembling**: Training and aggregating hundreds of deep models is computationally prohibitive in vision. In contrast, it is standard practice to ensemble hundreds of tree-based learners in tabular settings. **Our study provides detailed insights into the benefits (RQ2), efficiency (RQ4 and RQ5), and robustness (RQ6) of such ensembles.**
> - **The impact of resampling and cleaning**: As you noted, these preprocessing techniques are rarely effective in vision tasks. However, they remain practical and useful in tabular settings, especially when combined with ensemble methods. **We discuss their comparative performance and efficiency across paradigms in RQ1, RQ3, RQ4, and RQ5.**
> - **The role of data quality under limited data**: Unlike vision tasks that benefit from large curated datasets and pretrained models, tabular prediction models are often trained from scratch on smaller datasets. In such settings, the effect of noisy or incomplete data becomes much more critical. **We analyze this in both real-world settings (RQ1) and controlled experiments (RQ6 and RQ7).**
>
> In summary, while some surface-level observations may appear similar, the **underlying modeling assumptions, data characteristics, and effective solutions differ substantially** between the two domains. CLIMB aims to fill this gap by providing tabular-specific insights and benchmarking.
>
>
> [1] Lin, Tsung-Yi, et al. "Focal loss for dense object detection." ICCV, 2017.
> [2] Cui, Yin, et al. "Class-balanced loss based on effective number of samples." CVPR, 2019.
> [3] Johnson, Justin M., and Taghi M. Khoshgoftaar. "Survey on deep learning with class imbalance." Journal of Big Data 6.1 (2019): 1-54.
> [4] Ghosh, Kushankur, et al. "The class imbalance problem in deep learning." Machine Learning 113.7 (2024): 4845-4901.
> [5] Grinsztajn, Léo, Edouard Oyallon, and Gaël Varoquaux. "Why do tree-based models still outperform deep learning on typical tabular data?" NeurIPS, 2022.
>
>
> # Q2: Takeaway #3 (different metrics tell different stories) lacks novelty.
>
> > Takeaway #3 does not seem to offer a novel conclusion or observation specific to this paper. The idea that different evaluation metrics highlight distinct aspects of performance is well-established in the field. As a result, this takeaway does not provide new insights and appears to be more of a general principle rather than a unique finding of the paper.
>
> ## TL;DR
>
> **We agree this is a well-established idea and that's why we cited prior works when presenting it. We included it as a standalone takeaway due to its practical importance and frequent neglect in applied settings. It is more of a concise reminder supported by large-scale empirical evidence.**
>
> ## Response
>
> Thank you for the comment. **We agree that the idea behind Takeaway #3 is well-established. In fact, we explicitly referenced prior work discussing this issue in RQ3 to acknowledge its known importance.**
>
> We would also like to clarify that we do not present this as a novel finding, which is why we devoted less space to it compared to our other takeaways. That said, we chose to include it as a standalone takeaway because, despite its familiarity, it holds practical significance and is often overlooked in real-world applications.
>
> **Our goal is to reinforce this principle through large-scale empirical evidence and highlight concrete cases where the choice of metric can lead to different conclusions. We hope this serves as a concise but important reminder for practitioners, especially those who may rely solely on a single metric when evaluating performance under class imbalance.**
>
>
> ## Q3: Takeaway #5: Why combine noisy labels and class imbalance in a single experiment?
>
> > “In Takeaway #5, the rationale for combining the two issues—noisy data and class imbalance—into a single experiment is not entirely clear. These issues are typically treated as separate problems, each with its own set of specific techniques.”
>
> Thank you for the comment. We would like to clarify that we did **NOT** combine these factors into a single experiment. **As stated in Section 4.3 - Setup: “To ensure a fair comparison, each factor is introduced individually while keeping other factors unchanged.” This is also reflected in Figure 4, where each row corresponds to a separate factor to highlight the independence between factors.**
>
> To further clarify the intent, we have revised "noisy labels AND missing features" to "noisy labels OR missing features" in Takeaway #5, emphasizing that these factors were evaluated separately.
>
>
> ## Q4: About previous preprint of the IMBENS library
>
> > “The open-source library mentioned in the paper was previously introduced in the work by Imbens: Ensemble Class-imbalanced Learning in Python. Therefore, it is questionable whether it can be considered an original contribution of this study.”
>
> **Thank you for your careful review and for referencing the earlier version of our library!**
>
> **We would like to clarify that the work you cited "Imbens: Ensemble Class-imbalanced Learning in Python" is our own arXiv preprint describing the initial software design and motivation. That preprint has not been published in any peer-reviewed venue, and this NeurIPS submission represents the first formal publication of the benchmark and the library.**
>
> # Happy to have further discussion!
>
> **Thank you again for the thoughtful review. We’ve dedicated many efforts to form this rebuttal and will include them to enhance the paper’s quality. We hope our responses address your concerns and are happy to discuss if you have any further questions!**

---

> > ### Author Response · Authors · 2025-08-07
> > **Happy to have further discussion**
> >
> > Dear Reviewer G91d,
> >
> > Thank you for your acknowledgment. We would greatly appreciate it if you could briefly let us know whether our rebuttal has addressed your concerns, or if you have any further questions. We are eager to understand if any issues remain unresolved and are happy to further clarify.
> >
> > Best regards,
> > Authors

---

### Official Review · Reviewer_qi2z · 2025-06-30

**Rating:** 5
**Confidence:** 5

**Summary:**

In this paper, the authors introduce a benchmark for class-imbalanced learning for tabular data. To this end, they introduce an easy-to-use Python package which enables working with 29 class-imbalanced learning algorithms on 73 carefully selected tabular datasets. Furthermore, the authors provide a systematic comparison among the selected algorithms and provide interesting take-away messages.

**Dataset Code Accessibility:**

Yes

**Dataset Code Comments:**

The benchmark is provided as a Github repository. It appears that this repository has been around for a while.

**Ethical Considerations:**

No, there are no or only very minor ethics concerns

**Final Justification:**

I would like to thank the authors for the detailed responses to my concerns. I am especially glad that newer methods have been experimented with and I am surprised about the inferior performance of the deep learning based and other newer approaches in tabular data. Despite their inferior performances, I strongly believe that they should be included in the paper at least for the sake of completeness.

Having satisfied with the responses, I will increase my recommendation.

**Limitations Weaknesses:**

I am generally fond of the paper but I would like to raise the following concerns:

1. The only weakness I see with the paper is that the 29 class-imbalanced learning algorithms considered in the paper are rather outdated (mostly before 2010, with three exceptions in 2014, 2017 and 2020). Considering the fact that the literature has passed beyond the selected methods, one wonders whether the proposed benchmark will be of interest to the community.

2. The benchmark is limited to tabular data/problems. With suitable dataloaders, the benchmark should be easily extensible to other modalities (e.g., images, audio, ...) and many methods considered in the paper should be directly applicable (e.g., cost sensitive learning, resampling, ensemble..).

3. The results in the main paper and the take-away messages are based on an average over the 73 datasets. Due to space limitations and for the sake of brevity, the authors included dataset-dependent results in the appendix, which is fair. However, the main paper should discuss that there are datasets contradicting the take-away messages, hopefully with an explanation about why.

**Strengths Contributions:**

+ A well-prepared Python package.
+ Extensive benchmark with many algorithms and datasets.
+ A systematic comparison among the selected algorithms.
+ Helpful insights and take-away messages.

---

> ### Author Rebuttal · Authors · 2025-07-29
>
> **Thank you for the encouraging review and helpful feedback! We've conducted new experiments to address your concerns and updated the paper accordingly. Due to the no‑link policy, we can't include new figures or links, but we try our best to summarize the key insights in text.**
>
> **Below are our point‑by‑point responses. TL;DRs are provided before detailed explanations.**
>
>
> # Q1: Compare with more recent methods
>
> > “The 29 class-imbalanced learning algorithms considered in the paper are rather outdated (mostly before 2010, with three exceptions in 2014, 2017 and 2020).”
>
> ## TL;DR
>
> **We intentionally focused on widely adopted, representative CIL methods across key paradigms. While many are older, recent CIL methods have limited impact, especially in tabular data, where tree-based models still dominate. We further show that several CIL methods (e.g., SPE, RUSBoost) outperform modern GBDTs like XGBoost, LightGBM, and CatBoost. We’re happy to include newer methods if the reviewer has specific suggestions.**
>
> ## Response
>
> Thank you for your careful reading. Our goal in this benchmark is to provide a comprehensive and reproducible evaluation of **representative and influential CIL methods** across different paradigms (undersampling, oversampling, cost-sensitive learning, ensemble learning, etc.). We intentionally focused on methods that have been **widely adopted, cited, and used in practice, many of which remain state-of-the-practice even today.**
>
> In preparing this benchmark, we reviewed recent literature but found few recent CIL methods have gained comparable adoption or publication in top-tier venues, especially in the tabular domain. Many of the newer methods are designed for deep models or tailored to specific modalities (e.g., images, text), which fall outside the scope of our current benchmark.
>
> Recent studies [1,2] also support the continued dominance of **tree-based models** (e.g., XGBoost, LightGBM) for tabular data. To this end, we conducted additional experiments comparing our CIL methods with these strong GBDT baselines. We updated Figure 5 (model-to-model win rates) to include XGBoost, LightGBM, and CatBoost using the same ensemble size as other methods. Interestingly, several CIL methods (e.g.，SPE, RUSBoost, and SMOTEBagging) consistently outperform these SOTA GBDTs even with less-optimized scikit-learn base learner. This trend holds for AUPRC and macro-F1, while additional methods (e.g., BC, BRF, UnderBagging) also show advantage in Balanced Accuracy.
>
>
> |WinRate vs.|XGB-AP|XGB-F1|XGB-BACC|LGB-AP|LGB-F1|LGB-BACC|CAT-AP|CAT-F1|CAT-BACC|
> |---|---|---|---|---|---|---|---|---|---|
> |SPE|65.8%|67.1%|89.0%|58.2%|63.7%|84.2%|61.0%|67.8%|89.7%|
> |UBS|54.8%|52.1%|93.2%|53.4%|54.8%|93.2%|54.8%|58.9%|94.5%|
> |SMBA|45.2%|52.1%|57.5%|45.9%|57.5%|66.4%|50.0%|59.6%|73.3%|
>
> **That said, we truly appreciate the reviewer’s comment. If there are specific newer CIL methods you would like us to compare with, we would be glad to include them in our discussion.**
>
> [1] Grinsztajn, L., Oyallon, E., & Varoquaux, G. (2022). *Why do tree-based models still outperform deep learning on typical tabular data?* NeurIPS.
> [2] Shwartz-Ziv, R., & Armon, A. (2022). *Tabular data: Deep learning is not all you need.* Information Fusion.
>
>
> # Q2: Benchmark is limited to tabular data, could it extend to other modalities?
>
> > “The benchmark is limited to tabular data/problems. With suitable dataloaders, the benchmark should be easily extensible to other modalities (e.g., images, audio, ...)”
>
> ## TL;DR
> **Although CLIMB focuses on tabular data, it already includes datasets derived from other modalities (e.g., image and speech recognition) in tabularized form. But still, due to the unique challenges and architectures in non-tabular domains, we believe class imbalance should be benchmarked independently there, with CLIMB serving as a solid foundation for future extensions.**
>
>
> ## Response
> We agree with the reviewer’s insight that the benchmark framework could, in principle, be extended to other modalities. Although our current work focuses on tabular data, many of the core ideas underlying the evaluated CIL methods (such as resampling, reweighting, and ensembling) are general and have been applied in other domains including vision and language.
>
> In fact, our benchmark includes several datasets that originate from other modalities (e.g., image and speech recognition). As shown in **Appendix A.1, Table 3**, these datasets were transformed into tabular form via feature extractors applied to raw image or audio inputs. This demonstrates that our empirical insights are **not strictly limited to purely structured tabular data**, and can generalize to other modalities when their features are similarly represented.
>
> That said, we emphasize that **not all CIL methods designed for tabular data transfer directly to other modalities**. For example, ensembling hundreds of decision trees is efficient and effective in tabular domains, but ensembling large neural networks (common in vision or NLP) is computationally prohibitive. Moreover, modality-specific challenges (such as spatial coherence in images or temporal dynamics in audio) often require customized CIL techniques beyond those used in tabular learning.
>
> **For these reasons, we believe that class imbalance in other modalities should be studied and benchmarked independently**, using modality-appropriate datasets, models, and evaluation protocols. That said, we hope our findings in CLIMB can offer a strong foundation and valuable reference for future work on class-imbalance in broader machine learning contexts.
>
> # Q3: Discussions on contradicting datasets?
>
> > “the main paper should discuss that there are datasets contradicting the take-away messages, hopefully with an explanation about why.”
>
> ## TL;DR
>
> **We've intentionally phrased our takeaways to avoid overgeneralization as we acknowledge that a few datasets deviate from the overall trends. For example, random undersampling ensembles struggle on extremely imbalanced datasets, and cleaning methods can underperform on long-tailed multiclass distributions. We will add such dataset-level discussions in the revised paper to provide clearer context.**
>
> ## Response
>
> Thank you for the thoughtful suggestion and your understanding. We fully agree that it's important to acknowledge and interpret exceptions to the overall trends. **In fact, we have carefully phrased our takeaways to avoid overgeneralization. That said, we recognize that some datasets deviate from the average behavior.** Below we briefly discuss some notable cases:
>
> 1. **Some undersample-ensemble method fail on extremely imbalanced datasets** (e.g., *dis*, *satellite*): Although generally have great performance, random-undersampling-based ensemble methods (e.g., BRF, EasyEnsemble, and UBS) perform poorly on these datasets because they discard most majority-class samples, leading to insufficient information from training data and poor generalization. In contrast, methods like SPE and BC, which incorporate hard example mining, are more robust as they focus on retaining informative samples.
> 2. **Cleaning-based methods have trouble on long-tailed multiclass datasets** (e.g., *user-knowledge*, *allbp*):
>    Cleaning-based methods like Tomek Links, ENN, and RENN often underperform on such datasets. These methods are designed to remove noisy or borderline instances, but in long-tailed scenarios, multiple minority classes can be close to majority classes in feature space. This causes cleaning methods to over-remove minority samples, weakening the model’s ability to learn from rare classes.
>
> **We will include these dataset-level observations in the revised version of the paper to better contextualize our findings. We would be happy to further discuss any other specific examples you are interested in.**
>
>
> # Happy to have further discussion!
>
> **Thank you again for the thoughtful review. We’ve dedicated many efforts to form this rebuttal and will include them to enhance the paper’s quality. We hope our responses address your concerns and are happy to discuss if you have any further questions!**

---

> > ### Comment · Reviewer_qi2z · 2025-08-01
> > **Re: rebuttal**
> >
> > I would like to thank the authors for the detailed responses to my concerns. I am especially glad that newer methods have been experimented with and I am surprised about the inferior performance of the deep learning based and other newer approaches in tabular data. Despite their inferior performances, I strongly believe that they should be included in the paper at least for the sake of completeness.
> >
> > Having satisfied with the responses, I will increase my recommendation.

---

> > > ### Author Response · Authors · 2025-08-01
> > > **Great thanks!**
> > >
> > > Again, we thank you for your thoughtful review and encouraging feedback! It means a lot to us : ) We will incorporate new results in the paper. Thank you again for your support!
> > >
> > > Best wishes,
> > > Authors

---

### Official Review · Reviewer_fc3T · 2025-07-02

**Rating:** 5
**Confidence:** 4

**Summary:**

The paper proposes the **Cl**ass-**imb**alanced Learning Benchmark (CLIMB) for studying 29 imbalanced learning methods on 73 real-world datasets collected from the OpenML platform, which provides large-scale experiments to show the limitations of undersampling methods, the effectiveness of ensembles for undersampling methods, and the impact of the noisy labels and missing features.

**Additional Feedback:**

I am willing to increase my scores if the authors could
1. address most problems and give acceptable reasons for the rebutting my questions;
2. give a step-by-step reproducing scripts/examples to reproduce results.

**Dataset Code Accessibility:**

Partly

**Dataset Code Comments:**

1. The README document is understandable and well-written.
2. The datasets are accessible.
3. However, the paper's summarized tables (results) require time to reproduce.

**Ethical Considerations:**

No, there are no or only very minor ethics concerns

**Final Justification:**

This work provided the IMBENS Python package and enough experiments for assessing the existing methods for imbalanced tabular datasets. I would suggest this work to practitioners in the imbalanced learning field.

I look forward to seeing the subsequent follower extend the framework to the modern deep learning framework and advanced imbalanced methods.

**Limitations Weaknesses:**

1. Lack of relationship to existing datasets, benchmarks. While this work enhances the imbalanced-learn, this work did not compare with the previous work accepted by NeurIPS 2022 D&B Benchmark, Bank Account Fraud (BAF) [1]. Please provide the comparisons between your work and the BAF benchmark, such as different experimental settings, evaluation criteria, etc.
2. The category of *[Combination of over- and under-sampling methods](https://imbalanced-learn.org/stable/references/combine.html)* is missing from this paper. imbalanced-learn provides SMOTEENN and SMOTETomek for this category. Is there any reason for skipping this category?
3. Lack of the types of imbalance. [2] provided the other aspect of imbalanced settings beyond the imbalanced ratio, which included *long-tailed imbalance* and *step imbalance* for the multiclass classification problems. Please provide some preliminary (summary) results for the two types of imbalance from the existing datasets.
4. More introduction and intuitive insights of **Self-paced Ensemble (SPE)** [3]. SPE shows the most effective and efficient in Table 2 and Figures 3- 4. Could you provide more information about this method and possible reasons for its outperformance?
5. In Section 4.3 Robustness Analysis, the settings are not sensible to me. Why do we replace the missing values with the mean value instead of keeping the missing values? I consider that the tree model implemented in scikit-learn could handle the missing values. Using the mean value could result in a bias of features towards the majority. Could you provide the experiments for keeping the missing values or the reasons for this setting?
> Missing value: Given the missing ratio, we randomly mask corresponding number of values across all samples and features, replace them with the mean value observed for each respective feature.

6. Lack of variance of 5-fold performance. Please provide the standard deviation of the results if possible.

- [1] NeurIPS 2022 D&B, Sérgio Jesus et al., Turning the Tables: Biased, Imbalanced, Dynamic Tabular Datasets for ML Evaluation
- [2] NeurIPS 2019, Kaidi Cao et al., Learning Imbalanced Datasets with Label-Distribution-Aware Margin Loss
- [3] ICDE 2020, Zhining Liu et al., Self-paced Ensemble for Highly Imbalanced Massive Data Classification

**Strengths Contributions:**

This work contributes to an easy-to-use Python package--IMBENS to allow practitioners to study and apply the imbalance learning methods to real-world scenarios easily. IMBENS enhances the imbalanced-learn, which is developed by scikit-learn, by refactoring the codebase to promote advanced ensemble learning methods. I agree with the good potential impact on the research community and enterprises for further imbalanced learning.

The paper's presentation is good to me with detailed settings and clear categories of existing imbalanced learning methods. The experiments are well organized with detailed figures and tables to answer questions about the benefits and weaknesses of different methods.

---

> ### Author Rebuttal · Authors · 2025-07-29
>
> **Thank you for the encouraging review and helpful feedback! We've conducted new experiments to address your concerns and updated the paper accordingly. Due to the no‑link policy, we can't include new figures or links, but we try our best to summarize the key insights in text.**
>
> **Below are our point‑by‑point responses. TL;DRs are provided before detailed explanations.**
>
> # Q1: Comparison with BAF
>
> Thank you for highlighting this relevant work. **We have carefully reviewed the BAF benchmark and will include a discussion in the revised paper to clarify its relationship with our work. We summarize the key distinctions below:**
>
> |Aspect|CLIMB|BAF|
> |---|---|---|
> |**Goal**|Evaluate class-imbalanced learning (CIL) methods|Evaluate fairness under data bias and temporal shift|
> |**Task Scope**|73 real-world tabular classification tasks|Single fraud detection task with synthetic variants|
> |**Data Source**|Real public datasets (e.g., OpenML)|Synthetic data from CTGAN trained on private bank data|
> |**Bias/Imbalance**|Natural imbalance in real data|Controlled injection of group bias and imbalance|
> |**Protected Attributes**|Not considered|Yes (e.g., age groups)|
> |**Privacy Handling**|Not needed (public data)|Differential privacy via noise+GAN|
> |**Evaluation Setup**|5x2 cross-validation|Temporal split (6 months train, 2 months test)|
> |**Performance Metrics**|AUPRC, Macro-F1, Balanced Accuracy|TPR@5%FPR, FPR ratio (predictive equality)|
>
>
> # Q2: About SMOTEENN & SMOTETomek
>
> ## TL;DR
>
> We excluded SMOTEENN and SMOTETomek due to their higher computational cost, added complexity, and weaker performance compared to their individual components. **We now include them and find no clear benefit over simpler alternatives or other paradigms.**
>
> ## Response
>
> We initially excluded them because they are
>
> 1. **slower**: These methods are slower than using SMOTE or ENN/Tomek alone. Applying distance-based cleaning like ENN or Tomek after synthetic over-sampling can be slow due to the enlarged dataset and the need for nearest-neighbor computations.
> 2. **more complex**: These methods introduce additional hyperparameters from both SMOTE and ENN/Tomek, making them harder to tune and less attractive for practitioners seeking simple and efficient solutions.
> 3. **underperforming individual counterparts**: In our preliminary experiments, we found that these combined methods often did not outperform, and in many cases underperformed, than applying SMOTE, ENN, or Tomek alone. We suspect this is due to over-removal of informative borderline samples. After SMOTE introduces synthetic points, the data distribution near class boundaries becomes denser and more overlapping, which makes ENN or Tomek behave more aggressively in removing instances (please see "Compare sampler combining over- and under-sampling" in imblearn example gallery for an example). This can lead to the loss of valuable borderline information that would not be removed by ENN/Tomek alone, thereby hurting classifier performance.
>
> That said, we acknowledge the reviewer’s suggestion and have revised our main results (e.g., Table 2, Figure 3, and Figure 5) to include SMOTEENN and SMOTETomek. Below are representative comparisons from the updated Figure 5 (model-to-model win rate).
>
> **Comparison with individual counterparts:** Despite being more complex, SMOTEENN and SMOTETomek often perform worse than their individual components.
>
> |WinRate vs.|Tomek|ENN|SMOTE|
> |-|-|-|-|
> |SMOTEENN|50.7%|26.0%|44.5%|
> |SMOTETomek|39.7%|20.5%|41.8%|
>
> **Comparison with other CIL paradigms (averaged over CIL methods in each paradigm):** Other than against plain undersampling, combined sampling methods offer no consistent advantage over other paradigms.
>
> |Avg. WinRate vs. |Base|Undersample|Cleaning|Oversample|UnderEns|OverEns|CostSensitive|
> |-|-|-|-|-|-|-|-|
> |SMOTEENN|50.7%|87.3%|34.2%|49.2%|8.0%|21.6%|39.4%|
> |SMOTETomek|38.4%|84.3%|26.9%|42.6%|8.0%|14.4%|32.9%|
>
> # Q3: Why not explore long-tailed and step imbalance types?
>
> ## TL;DR
> We agree that long-tailed and step imbalance are useful paradigms when **constructing synthetic imbalanced datasets**. However, in our benchmark, all datasets (both binary and multiclass) are **naturally imbalanced**, and we aim to study methods under real-world imbalance **without artificial manipulation**.
>
> ## Response
>
> Unlike prior work that simulates imbalance via long-tailed or step distributions, CLIMB uses only naturally imbalanced datasets without artificial perturbation. Our goal is to provide practical insights into CIL on real-world data, where adding synthetic imbalance may distort original distributions and reduce relevance. Therefore, we believe that evaluating CIL methods directly on naturally imbalanced datasets (as they exist) is a more faithful and meaningful reflection of real-world applications.
>
> # Q4: About Self-Paced Ensemble (SPE) and its advantage?
>
> Thank you for the suggestion. We will expand our discussion of SPE in the paper. Briefly, its strong performance stems from several factors:
>
> 1. **Hard example mining**: SPE retains hard-to-classify samples during undersampling, which improves decision boundaries.
> 2. **Noise robustness**: Its hardness harmonization mechanism balances informativeness and noise, avoiding overly noisy samples.
> 3. **Self-paced learning**: Inspired by curriculum learning, SPE introduces samples progressively from easy to hard, stabilizing training.
> 4. **Efficiency**: As an undersampling ensemble, SPE trains on fewer samples per model, making it more efficient than oversampling or boosting.
>
> # Q5: Why impute missing values with mean?
>
> ## TL;DR
> **While decision trees can handle missing values, many CIL methods like SMOTE, SMOTEENN, and SMOTETomek cannot, as they rely on nearest neighbors or interpolation which require complete data.**
>
> ## Response
> Although scikit-learn’s decision trees support missing values, **most CIL methods in our benchmark do not**. Techniques like SMOTE, SMOTEENN, and SMOTETomek rely on operations that require fully observed feature vectors, such as neighbor search or interpolation.
>
> To ensure **compatibility and fairness**, we apply simple mean imputation as a model-agnostic and reproducible approach across all datasets. While more advanced imputation methods exist, we prioritize consistency in this benchmark setting.
>
> # Q6: Standard deviation
>
> We will include standard deviations in the updated appendix, based on the existing dataset-level results reported in Tables 5–7.
>
> # Feedback 2: step-by-step reproducing scripts/examples
>
> Thank you for the helpful suggestion. We have prepared **clean and documented Python scripts and notebooks** that fully reproduce our reported results, including code blocks, comments, output logs, and config files (e.g., seeds, metrics). However, due to this year’s rebuttal policy **prohibiting links or repository updates**, we cannot share them at this stage unless explicitly allowed by the AC.
>
> We summarize below the core structure and steps of the reproducibility scripts:
>
> 1. **Load datasets**
>    - Use `load_datasets(directory="datasets", type="all")` to load all preprocessed datasets and accompanying statistics.
> 2. **Define evaluation metrics and CIL methods**
>    - Specify metrics such as `f1`, `bacc`, `ap`, and define resampling-based samplers (`SMOTE`, `ENN`, etc.) and ensemble classifiers (`SPE`, `RUSBoost`, etc.).
> 3. **Classifier construction utilities**
>    - Use helper functions like `get_clfs_resample`, `get_clfs_ensembles`, and `get_clf` to build models with desired resampling strategies or ensemble frameworks.
> 4. **Hyperparameter search space definition**
>    - All tunable hyperparameters (e.g., `n_neighbors`, `learning_rate`, `threshold_cleaning`) are defined in the `search_params` dictionary.
> 5. **Search space generator**
>    - The `get_param_space_func` function translates `search_params` into trial-based suggestion functions compatible with Optuna.
> 6. **Model optimizer**
>    - The `optimize_hyperparams_with_optuna` function runs Optuna-based tuning with early stopping and returns the best parameters and performance.
>    - **Outputs written to:**
>      - `results/searched.csv`: evaluation scores for each trial.
>      - `results/params/best_params.json`: best parameters for each dataset-method pair.
> 7. **Cross-validation configuration**
>    - Set via `trial_kwargs`, defining a 5-fold cross-validation repeated once, using macro-averaged metrics.
> 8. **Execution loop**
>    - Iterate through each dataset and method to run tuning and evaluation.
>    - Results are logged incrementally into:
>      - `results/searched.csv`
>      - `results/params/best_params.json`
>
> We hope this overview demonstrates the effort we've made to ensure reproducibility. We look forward to sharing the full scripts upon acceptance or earlier if permitted.
>
> # Happy to have further discussion!
>
> **Thank you again for the thoughtful review. We’ve dedicated many efforts to get the new results and will include them to enhance the paper’s quality. We hope our responses address your concerns and are happy to discuss if you have any further questions!**

---

> > ### Comment · Reviewer_fc3T · 2025-08-03
> >
> > > Q3: Why not explore long-tailed and step imbalance types?
> > > We agree that long-tailed and step imbalance are useful paradigms when constructing synthetic imbalanced datasets. However, in our benchmark, all datasets (both binary and multiclass) are naturally imbalanced, and we aim to study methods under real-world imbalance without artificial manipulation.
> >
> > While the authors stand on the *naturally imbalanced*, I believe the step imbalance type sometimes occurs in the real world when we manipulatively merge or split classes. I still suggest the authors add it as a preliminary investigation in the future version.
> >
> > The authors have resolved my issues. I have increased my score. Thanks.

---

> > > ### Author Response · Authors · 2025-08-03
> > > **Great thanks!**
> > >
> > > Thank you very much for your encouraging feedback! It means a lot to us : ) It makes sense that step imbalance may occur in the real world when we manipulatively merge or split classes, we will add relevant results and discussion in the revision. Thank you again for raising the score!
> > >
> > > Best wishes,
> > > Authors

---

### Official Review · Reviewer_FBFT · 2025-07-04

**Rating:** 4
**Confidence:** 5

**Summary:**

The authors introduce CLIMB, a benchmark for class imbalanced learning on tabular data, and IMBENS, an open source code-base for imbalanced learning. CLIMB consists of 73 imbalanced tabular datasets with an imbalance ratio of >= 2, and evaluates 30 imbalanced learning methods, including ensemble methods such as undersample ensembles. Overall, undersample ensembles perform best and have strong runtime properties compared to oversample ensembles. In particular, Self-Paced Ensembles outperform all methods on the AP metric.

**Additional Feedback:**

- Figure 5 would be more informative if it was sorted by average win-rate. Currently it is very hard to extract meaning out of.


Overall Comments:

This is great work and is very extensive. It seems to tick nearly all the boxes for a well implemented benchmark and evaluation. Kudos to the authors! With that said, it has major blind spots as I mention earlier in my review that I cannot overlook. Benchmarks must be anchored by strong, reliable, and practically useful baselines. The absence of XGBoost, LightGBM, and CatBoost from the paper is striking. Without justification for why these methods are excluded, I can't recommend acceptance.

**Dataset Code Accessibility:**

Yes

**Dataset Code Comments:**

Code-base is high quality, MIT licensed. Datasets are available on OpenML.

**Ethical Comments:**

No ethical concerns.

**Ethical Considerations:**

No, there are no or only very minor ethics concerns

**Final Justification:**

The paper is of high quality with many methods incorporated into the benchmark. My primary concern was the lack of mention or comparison with GBDT methods. This has since been addressed by the reviewers in the rebuttal, where they both compared to these methods as well as in the case of CatBoost used it as a base model to show how the techniques in the paper can further improve CatBoost. While I wish that such details had been present in the original paper, ultimately I believe their inclusion as mentioned in the rebuttal is sufficient for me to lean towards acceptance.

**Limitations Weaknesses:**

- **Major**: No mention of XGBoost, LightGBM, or CatBoost at any point.

If the goal is to analyze which techniques are state of the art, why are the state of the art tree based GBDT methods not evaluated or mentioned? Currently the base model used in the paper seems to be a simple decision tree / random forest, which is far outperformed by the previously mentioned methods. If the take-aways from the paper are to have practical relevance, they need to be compared against these or combined with these baselines. Currently, I do not know if any of the methods benchmarked in the paper would beat the default CatBoost configuration. This seems incredibly important, and I struggle to understand why this has not been done.

- No evaluation of any deep learning / neural network method at any point. Recent strong tabular deep learning methods include TabM, RealMLP, and ModernNCA.
- No evaluation of the log loss or roc auc metrics, which would avoid calibration issues skewing results.

**Strengths Contributions:**

- Code-base with a detailed readme, 370 GitHub stars, unit tests, and documentation.
- Many methods (29!) implemented for imbalanced learning.
- Detailed performance vs runtime analysis
- Detailed method comparison with ranks and multiple evaluation metrics
- Figure 5 with pair-wise win-rates
- 73 datasets with 5-fold cross-validation
- Robustness analysis ablation with injecting missing values

---

> ### Author Rebuttal · Authors · 2025-07-29
>
> **We are grateful for the encouraging review and constructive feedback! With our best efforts, we conducted numerous additional experiments and analyses to address your concerns. The manuscript has been updated accordingly. However, please understand that due to this year’s no‑link policy, we are unable to include new figures or links to new results/updated pdf. We try our best to summarize the key insights in text.**
>
> **Below we provide the point‑by‑point responses. For your convenience, we provied TL;DR summaries before lengthy detailed explanations to help you quickly grasp our main points.**
>
> # Q1: Why are XGBoost, LightGBM, and CatBoost not included?
>
> > “No mention of XGBoost, LightGBM, or CatBoost at any point. … Currently the base model used in the paper seems to be a simple decision tree / random forest … I do not know if any of the methods benchmarked in the paper would beat the default CatBoost configuration. … I struggle to understand why this has not been done.”
>
> **Thank you for raising this important point! In fact, we had extensive internal discussions on this as well.**
>
> ## TL;DR
>
> **We excluded XGBoost, LightGBM, and CatBoost from the main benchmark to ensure fair comparison among CIL methods using consistent base learners. That being said, there are ensemble-based CIL methods (e.g., SPE, RUSBoost) can outperform these GBDTs even with weaker scikit-learn tree base learners. We further conducted additional experiments using CatBoost as the base model for all CIL methods, results showing that the benchmarked CIL methods further improves CatBoost on class-imbalanced datasets.**
>
> ## 1. Why didn't include them?
> We decided not to include XGBoost, LightGBM, or CatBoost because their base tree learners are optimized in unique ways (e.g., with Hessians, histogram-based splitting, additional regularizations), making them fundamentally different from the scikit-learn decision tree used across our CIL methods. **This makes direct comparison potentially unfair and shifts focus away from the CIL techniques themselves.**
>
> ## 2. Any of the methods can beat CatBoost or other GBMs?
>
> Yes. We udpated Figure 5 (model-to-model win rate) to include XGBoost, LightGBM, and CatBoost results using same ensemble size as other ensemble methods. **SPE, RUSBoost, and SMOTEBagging generally outperforms (with over 50% win rate) all of them even with less-optimized sklearn base tree learner.** This is consistent for AUPRC and macro F1 score. More CIL methods (e.g., BC, BRF, UnderBagging) are showing advantage over XGB/LGB/CAB in Balanced Accuracy. We attach part of the updated Figure 5 results below.
>
> |WinRate vs.|XGB-AP|XGB-F1|XGB-BACC|LGB-AP|LGB-F1|LGB-BACC|CAT-AP|CAT-F1|CAT-BACC|
> |---|---|---|---|---|---|---|---|---|---|
> |SPE|65.8%|67.1%|89.0%|58.2%|63.7%|84.2%|61.0%|67.8%|89.7%|
> |UBS|54.8%|52.1%|93.2%|53.4%|54.8%|93.2%|54.8%|58.9%|94.5%|
> |SMBA|45.2%|52.1%|57.5%|45.9%|57.5%|66.4%|50.0%|59.6%|73.3%|
>
>
> ## 3. Using CatBoost as a base learner.
>
> Inspired by your feedback, we think it makes more sense to use those advanced GBMs as base models for CIL. We conducted additional experiments using CatBoost as the base learner and found that **CIL methods can be used on top of CatBoost and further boost its performance on class-imbalanced datasets.**
>
> Below we summarize the averaged performance of representative CIL methods using CatBoost as the base learner. For each CIL category, we report the top two performing methods (average scores across all datasets). The numbers in parentheses indicate the relative gain over vanilla CatBoost.
>
> |Metric|Base|RUS|IHT|ENN|NCR|ROS|SMT|SPE|UBA|OBA|SMBA|AdaC|AdaBS|
> |-|----|---|---|---|---|---|---|---|---|---|----|----|-----|
> |AP|52.4|48.2 (-4.2)|44.5 (-7.8)|54.6 (+2.3)|54.7 (+2.3)|53.9 (+1.6)|54.3 (+1.9)|57.8 (+5.4)|50.9 (-1.5)|56.4 (+4.0)|56.4 (+4.0)|56.3 (+4.0)|56.1 (+3.7)|
> |F1|76.8|73.9 (-3.0)|70.5 (-6.4)|77.9 (+1.0)|78.1 (+1.3)|77.9 (+1.1)|78.5 (+1.6)|79.3 (+2.4)|75.9 (-0.9)|79.7 (+2.8)|79.7 (+2.9)|78.7 (+1.8)|79.0 (+2.1)|
> |BAC|75.4|82.1 (+6.7)|80.3 (+4.9)|79.0 (+3.6)|78.9 (+3.4)|81.8 (+6.4)|81.4 (+6.0)|82.5 (+7.0)|83.0 (+7.6)|81.9 (+6.4)|81.6 (+6.2)|77.5 (+2.0)|79.1 (+3.7)|
>
>
> # Q2: Why are no deep tabular methods evaluated?
>
> > “No evaluation of any deep learning / neural network method at any point. Recent strong tabular deep learning methods include TabM, RealMLP, and ModernNCA.”
>
> ## TL;DR:
> **Deep tabular models are not included due to their high complexity, sensitivity to hyperparameters, and limited focus on class imbalance. Our benchmark emphasizes reproducibility and broad applicability, which is more feasible with tree-based methods.**
>
> ## Response:
>
> Thank you for the suggestion. We fully acknowledge the recent progress in tabular deep learning. However, our primary focus in this work is on **class-imbalanced** learning, which is typically not the main objective of these deep models. We will include a discussion of this in the revised paper.
>
> While we agree that deep models like TabM, RealMLP, and ModernNCA could, in principle, serve as base learners within our framework, several practical challenges prevent their inclusion at this stage:
>
> 1. Deep models typically have many sensitive hyper-parameters (at least, learning rate/scheduler, batch size, number of epoches, model width/depth, etc.) that significantly affect performance and require extensive tuning for each dataset.
> 2. Due to (1), combined with the complexity and training cost of deep models, make it difficult to include them in our benchmark. In particular, our setting requires tuning each method across 73 datasets with multi-fold cross-validation, which is computationally prohibitive for deep models.
>
> **For the above reasons, and considering that tree-based methods are still the go-to choice in practice for tabular data, we chose to scope this benchmark around tree-based methods for now.**
>
> # Q3: Why not using log loss or ROC AUC?
>
> > “No evaluation of the log loss or roc auc metrics, which would avoid calibration issues skewing results.”
>
> ## TL;DR
>
> **(i) The metric AP (Area under Precision-Recall curve) is also an AUC metric that avoids calibration (thresholding) issues.
> (ii) Just like classification Accuracy, ROC AUC and log loss can be dominated by the majority class(es), thus can be misleading under class imbalance. Compared to them, the metrics we used are more informative and reliable under class imbalance.**
>
>
> ## Response
>
> Thank you for the suggestion. We understand your concern on the calibration issues of F1 and Balanced Accuracy scores. That's why we also include AP (Area under Precision-Recall curve) as a major metric. Like ROC AUC, AUPRC is also an AUC metric that consider all possible thresholding over the predicted probabilities, and thus avoids calibration issues.
>
> More importantly, we note that the three metrics were chosen because they are known to be **more informative and reliable under class imbalance** [1,2].
> In particular:
> - AUPRC avoids calibration issues and focuses directly on the trade-off between precision and recall for the positive (minority) class, without being inflated by true negatives.
> - Macro F1 and Balanced Accuracy treats all classes equally and penalizes poor performance on the minority class.
>
> While ROC AUC and log loss are useful in general contexts, they are less aligned with the specific goals of class-imbalanced learning:
> - ROC AUC includes the true negative rate, which can dominate in imbalanced datasets, leading to misleadingly high scores [1].
> - Log loss reflects probabilistic calibration but is often driven by the majority class, thus failing to highlight minority-class performance.
>
> That said, if the reviewer feels it would strengthen the paper, we are happy to include ROC AUC and log loss results in the appendix for completeness. However, we believe the current metrics more directly and accurately assess class-imbalanced learning effectiveness, particularly with regard to minority-class behavior.
>
> > [1] Saito, Takaya, and Marc Rehmsmeier. "The precision-recall plot is more informative than the ROC plot when evaluating binary classifiers on imbalanced datasets." PloS one 10.3 (2015): e0118432.
> > [2] Johnson J M, Khoshgoftaar T M. Survey on deep learning with class imbalance[J]. Journal of big data, 2019, 6(1): 1-54.
>
> # Q4: Improving the presentation of Figure 5.
>
> > Figure 5 would be more informative if it was sorted by average win-rate. Currently it is very hard to extract meaning out of.
>
> Thank you for the helpful suggestion. In the current version, the row and column order of Figure 5 is intentionally aligned with Table 2, i.e., grouping methods by their respective CIL paradigms (e.g., undersampling, oversampling, cost-sensitive, ensemble) to facilitate intra- and inter-paradigm comparisons.
>
> To improve readability while preserving this structure, we will enhance Figure 5 by adding an additional layer of paradigm labels to the row and column headers. This should help readers quickly identify and locate methods within each paradigm and better interpret intra-/inter-paradigm win-rate relationships.
>
> # Happy to have further discussion!
>
> **Thank you again for the thoughtful review. We’ve dedicated many efforts to get the new results and will include them to enhance the paper’s quality. We hope our responses address your concerns and are happy to discuss if you have any further questions!**

---

> > ### Comment · Reviewer_FBFT · 2025-08-01
> >
> > Thank you for the detailed rebuttal!
> >
> > ## Q1
> >
> > > This makes direct comparison potentially unfair and shifts focus away from the CIL techniques themselves.
> >
> > I understand the perspective. In my view, methods that augment a model's behavior should prioritize being incorporated into the strongest base models, as their strengths may be different depending on the model they are augmenting, and so what is truly SOTA depends on this combination. I've seen many benchmarks in the past that have exclusively used weak models as their base to augment on, which then did not transfer their insights when applied to stronger base models, making the study's impact negligible in practice.
> >
> > With that said, I am very pleased to see that the authors incorporate the techniques into CatBoost and demonstrate an improved over the baseline CatBoost algorithm. I would have liked to learn if the augmented CatBoost outperformed the previous strongest solution in the benchmark, but I don't think the authors have shared that information in the rebuttal.
> >
> > I am also glad the authors add the 3 GBDT methods as baselines. This strengthens the relevance of the benchmark, as these are often the go-to solutions to tabular problems, especially for those who are unfamiliar with the particular suite of methods focused on in this paper. So it is important to convince these practitioners that they should adopt a new method as it outperforms the one they are currently using.
> >
> > ## Q2
> >
> > This is reasonable. I agree that traditionally deep learning methods have been painful to benchmark compared to tree methods. However, given the rapid progress of deep learning methods for tabular data in the past year (including TabPFNv2 and TabICL alongside those previously mentioned), it will probably be very important to begin incorporating them when possible into benchmarks going forward.
> >
> > ## Q3
> >
> > Makes sense. The primary thought I had was that these additional metrics could identify potential issues / low hanging fruit in the benchmark, such as a simple lack of a calibrated decision threshold for a model that gets poor balanced accuracy performance but strong log loss performance. But I don't think they are strictly necessary.
> >
> > ## Q4
> >
> > Sounds reasonable, thanks!
> >
> > ## Conclusion
> >
> > Thanks again for the detailed rebuttal! My primary concern regarding the lack of GBDT methods has been mostly addressed by the authors, and thus I have increased my score.

---

> > > ### Author Response · Authors · 2025-08-01
> > > **Great thanks!**
> > >
> > > Again, we sincerely appreciate your thoughtful review and encouraging feedback! It means a lot to us : ) Following up on Q1, the augmented CatBoost did slightly outperform (on AUPRC and ACC, tied on F1) the previous strongest solution in the benchmark, we will note that in the paper as well. Thank you again for your support!
> > >
> > > Best wishes,
> > > Authors

---

### Decision · Program_Chairs · 2025-09-18

**Decision:**

Accept (poster)

**Comment:**

This paper presents CLIMB, a comprehensive benchmark for class-imbalanced learning (CIL) on tabular data. The benchmark covers 73 real-world datasets and 29 CIL algorithms, and is released as an open-source Python package with unified APIs and strong reproducibility support. The work provides systematic evaluations and practical insights into method accuracy, efficiency, and robustness.

The paper received scores of 4, 5, 5, and 4. Reviewers praised the breadth and quality of the benchmark, the clarity of presentation, and the high potential impact of the released package. Concerns were raised about missing baselines (notably GBDT methods and modern deep tabular approaches), limited comparisons to related benchmarks, and the generality of some takeaways. The rebuttal addressed these issues by including GBDT and newer baselines, which reviewers found satisfactory. Overall, the contribution is viewed as timely, valuable, and impactful for the community, and I recommend acceptance.